# Unique structure and function of viral rhodopsins

Dmitry Bratanov[1,2,3,18], Kirill Kovalev[1,2,3,4,5,6,18], Jan-Philipp Machtens [6,7], Roman Astashkin[3,4], Igor Chizhov[8], Dmytro Soloviov [4,9,10,11], Dmytro Volkov[1,2,4], Vitaly Polovinkin[1,3], Dmitrii Zabelskii [1,2,4], Thomas Mager[12], Ivan Gushchin [4], Tatyana Rokitskaya [13], Yuri Antonenko[13], Alexey Alekseev[1,2,4,5], Vitaly Shevchenko[1,2,4,5], Natalya Yutin[14], Riccardo Rosselli [15], Christian Baeken[1,2], Valentin Borshchevskiy[1,2,4], Gleb Bourenkov [16], Alexander Popov[17], Taras Balandin[1,2], Georg Büldt[4], Dietmar J. Manstein[8], Francisco Rodriguez-Valera [15], Christoph Fahlke[4,6], Ernst Bamberg[4,12], Eugene Koonin [14] & Valentin Gordeliy[1,2,3,4]*

Recently, two groups of rhodopsin genes were identified in large double-stranded DNA viruses. The structure and function of viral rhodopsins are unknown. We present functional characterization and high-resolution structure of an Organic Lake Phycodnavirus rhodopsin II (OLPVRII) of group 2. It forms a pentamer, with a symmetrical, bottle-like central channel with the narrow vestibule in the cytoplasmic part covered by a ring of 5 arginines, whereas 5 phenylalanines form a hydrophobic barrier in its exit. The proton donor E42 is placed in the helix B. The structure is unique among the known rhodopsins. Structural and functional data and molecular dynamics suggest that OLPVRII might be a light-gated pentameric ion channel analogous to pentameric ligand-gated ion channels, however, future patch clamp experiments should prove this directly. The data shed light on a fundamentally distinct branch of rhodopsins and may contribute to the understanding of virus-host interactions in ecologically important marine protists.

[1] Institute of Complex Systems (ICS), ICS-6: Structural Biochemistry, Forschungszentrum Jülich, Jülich, Germany. [2] JuStruct: Jülich Center for Structural Biology, Forschungszentrum Jülich, Jülich, Germany. [3] Institut de Biologie Structurale J.-P. Ebel, Université Grenoble Alpes-CEA-CNRS, Grenoble, France. [4] Moscow Institute of Physics and Technology, Dolgoprudniy, Russia. [5] Institute of Crystallography, University of Aachen (RWTH), Aachen, Germany. [6] Institute of Complex Systems (ICS), ICS-4: Zelluläre Biophysik, Forschungszentrum Jülich, Jülich, Germany. [7] Department of Molecular Pharmacology, University of Aachen (RWTH), Aachen, Germany. [8] Institute for Biophysical Chemistry, Hannover Medical School, Hannover, Germany. [9] Joint Institute for Nuclear Research, Dubna, Russia. [10] Taras Shevchenko National University of Kyiv, Kyiv, Ukraine. [11] Institute for Safety Problems of Nuclear Power Plants NAS of Ukraine, Chornobyl, Kyiv Region, Kyiv, Ukraine. [12] Max Planck Institute of Biophysics, Frankfurt am Main, Germany. [13] Belozersky Institute of Physico-Chemical Biology, Lomonosov Moscow State University, Moscow, Russia. [14] National Center for Biotechnology Information, National Library of Medicine, National Institutes of Health, Bethesda, MD, USA. [15] Evolutionary Genomics Group, Departamento de Producción Vegetal y Microbiología, Universidad Miguel Hernández, San Juan de Alicante, Spain. [16] European Molecular Biology Laboratory, Hamburg unit c/o DESY, Hamburg, Germany. [17] European Synchrotron Radiation Facility, Grenoble, France. [18]These authors contributed equally: Dmitry Bratanov, Kirill Kovalev. *email: valentin.gordeliy@ibs.fr

Microbial rhodopsins that are extremely distantly related to animal visual rhodopsins comprise an expansive family of seven transmembrane proteins that contain a covalently attached cofactor, retinal[1,2]. Upon absorption of a photon, retinal isomerizes triggering a series of structural transformations that correlate with functional and spectral changes and are known as the photocycle[3–5]. Microbial rhodopsins are currently considered to be universal and the most abundant biological light energy transducers. Before the year 2000, only microbial rhodopsins from halophilic archaea have been known. A 2000 metagenomic study resulted in the discovery of a rhodopsin gene in marine Proteobacteria that was, accordingly, named proteorhodopsin (PR)[6]. Since 2000, thousands of microbial rhodopsins have been identified, in all three domains of life (bacteria, archaea and eukaryota) as well as in large viruses[3]. The renaissance of rhodopsins as a research field has culminated in the development of optogenetics, the revolutionary method for controlling cell behavior in vivo in which microbial rhodopsins play the key role[7,8].

Several rhodopsins with unexpected functions have been discovered and characterized recently. Among the members of this family are light-driven proton, anion and cation pumps, light-gated anion and cation channels, and photoreceptors[4,5,9,10]. Recently, rhodopsins that function as inward proton pumps have been discovered[11,12]. Genomic and metagenomic studies have dramatically expanded the collection of rhodopsin sequences, some of which have been identified in unexpected organisms and habitats, for example, sodium-pumping rhodopsins (NaRs) in Flavobacteria[13,14], and the wide spread and importance of PR-based phototrophy in the marine environment[15] have become evident. However, most of microbial rhodopsin diversity remains experimentally uncharacterized.

Computational analysis of proteins encoded by Nucleocytoplasmic Large DNA Viruses (NCLDV) of eukaryotes has led to the discovery of genes encoding PR homologs in Organic Lake Phycodnavirus and Phaeocystis globosa virus which belong to the extended Mimiviridae family[16]. Subsequently, a number of additional rhodopsins have been identified in other NCLDV[17,18] (Supplementary Fig. 1). These viral rhodopsins show only distant (albeit statistically significant) sequence similarity to microbial rhodopsins with known functions. Phylogenetic analysis shows that viral rhodopsins are monophyletic and split into two distinct branches[16] (Supplementary Figs. 2, 3). The structure, function, and role of the viral rhodopsins in the infection of the host protists remain unknown. Given the distant relationship between the viral rhodopsins and the rest of the microbial rhodopsin superfamily, the former could be expected to have unique properties. Furthermore, taking into account that P. globosa is a dominant component of marine phytoplankton whose population dynamics is substantially affected by viruses[19], viral rhodopsins could be among the major players in ocean ecology. These considerations prompted us to characterize viral rhodopsins experimentally.

Here we present a structural and functional characterization of Organic Lake Phycodnavirus rhodopsin II (OLPVRII), a representative of the second group of viral rhodopsins. The study reveals remarkable differences between viral rhodopsins and all other known microbial rhodopsins. Examination of the crystal structure, functional characterization and molecular dynamics (MD) simulations suggest that OLPVRII is a pentameric light-gated channel that is functionally analogous to well-studied pentameric ligand-gated ion channels playing crucial roles in many cellular processes[20]. The functional characteristics of OLPVRII described here may make this group of proteins promising candidates for the development of new optogenetic tools. However, channel ability of viral rhodopsins needs to be demonstrated experimentally.

## Results

**Overall OLPVRII structure.** OLPVRII was crystallized using the *in meso* approach similar to that used in our previous works[10,14,21]. Typically for *in meso* crystallization, type I pyramid-shape crystals up to 50 μm in size (Supplementary Fig. 4a) appeared within 3 months. The crystals diffracted up to 1.9 Å, and the structure of OLPVRII was solved at 1.9 Å. The space group of the crystals was determined to be $P2_1$, with the cell lattice parameters of $a = 79.58$ Å, $b = 99.70$ Å, $c = 82.96$ Å, $\alpha = \gamma = 90°$ and $\beta = 116.95°$. Crystal packing and examples of the electron density maps are presented in Supplementary Fig. 4b-e.

OLPVRII asymmetric unit cell contains five molecules of the protein organized in a pentamer with a bottle-like pore in the centre that has a narrow neck in the cytoplasmic part (Fig. 1a, d). Several lipid moieties with disordered polar heads are observed around the pentamer. In addition, a well-ordered molecule of monoolein (MO), host molecule of the crystallization matrix, is observed between the D helix of one protomer and the G′ helix of the neighboring pentamer. MO is buried deep inside the hydrophobic groove formed by Phe197, Ile200, Tyr201, Phe205, Phe210 of one monomer and Phe103′ of the other monomer (Supplementary Fig. 5). The location of the MO polar head is stabilized by hydrogen bonding with the main chain of Phe210 and via Wat20 with Ala204 and Ile 209 at the C-terminus.

**Organization of OLPVRII pentamer.** OLPVRII pentamer is stabilized by a dense net of hydrogen bonds. The most intensive contacts between protein molecules in the pentamer are located on the cytoplasmic side of the protein, where the helix A of one protomer is interposed into the cleft between helices A′ and B′ of other protomer (Fig. 1b). The existence of the entire network is conditioned on the presence of two highly ordered water molecules. The first one (Wat12) coordinates hydrogen bonding of the side chains of His37 and Asn40 of one protomer and Trp203′ and Glu26′ of the other (Fig. 1c, f). Asn40 is stabilized also by a hydrogen bond with the backbone oxygen of Ile22′. The second water molecule coordinates the interactions between Arg36 and backbone oxygens of Ala27 and Thr30 to further stabilize the position of Arg29′ side chain of the neighbor protein molecule. At the center of the pentamer, oxygens of Leu28 and Arg29 side chains of all five protomers are also bound by strong hydrogen bonds mediated by five water molecules (Wat13). Moreover, Glu26′ of the second protomer strongly interacts with Arg36 from the same protomer providing a connection between the clusters of water-mediated hydrogen bonds. Thus, the following amino acids play the key role in the formation of the pentamer: Glu26, Arg36, His37, Asn40, Trp203, and these amino acids are highly conserved within group II of viral rhodopsins (Supplementary Fig. 1), but not in other rhodopsins. In addition, the distorted C-terminus of one protomer interacts with the poorly ordered C−D loop of the neighboring protomer forming several hydrogen bonds between their backbone atoms and thus stabilizing the pentamer (Fig. 1b, e).

To prove that the observed pentameric assembly is not an artifact of crystallization, we performed crosslinking experiments with wild-type protein. Using size exclusion chromatography, the protein was separated into two fractions corresponding to monomers and pentamers (Supplementary Fig. 6a). Crosslinking experiments clearly confirm that multimeric fractions appearing on the size exclusion chromatography comprise pentamers (Supplementary Fig. 6b, lane 4). Moreover, we reconstituted the monomers of solubilized wild-type protein into liposomes, and crosslinking experiments showed that in these liposomes OLPVRII protein is in the pentameric form (Supplementary Fig. 6b, lane 5). Thus, the pentamers retain their oligomeric form

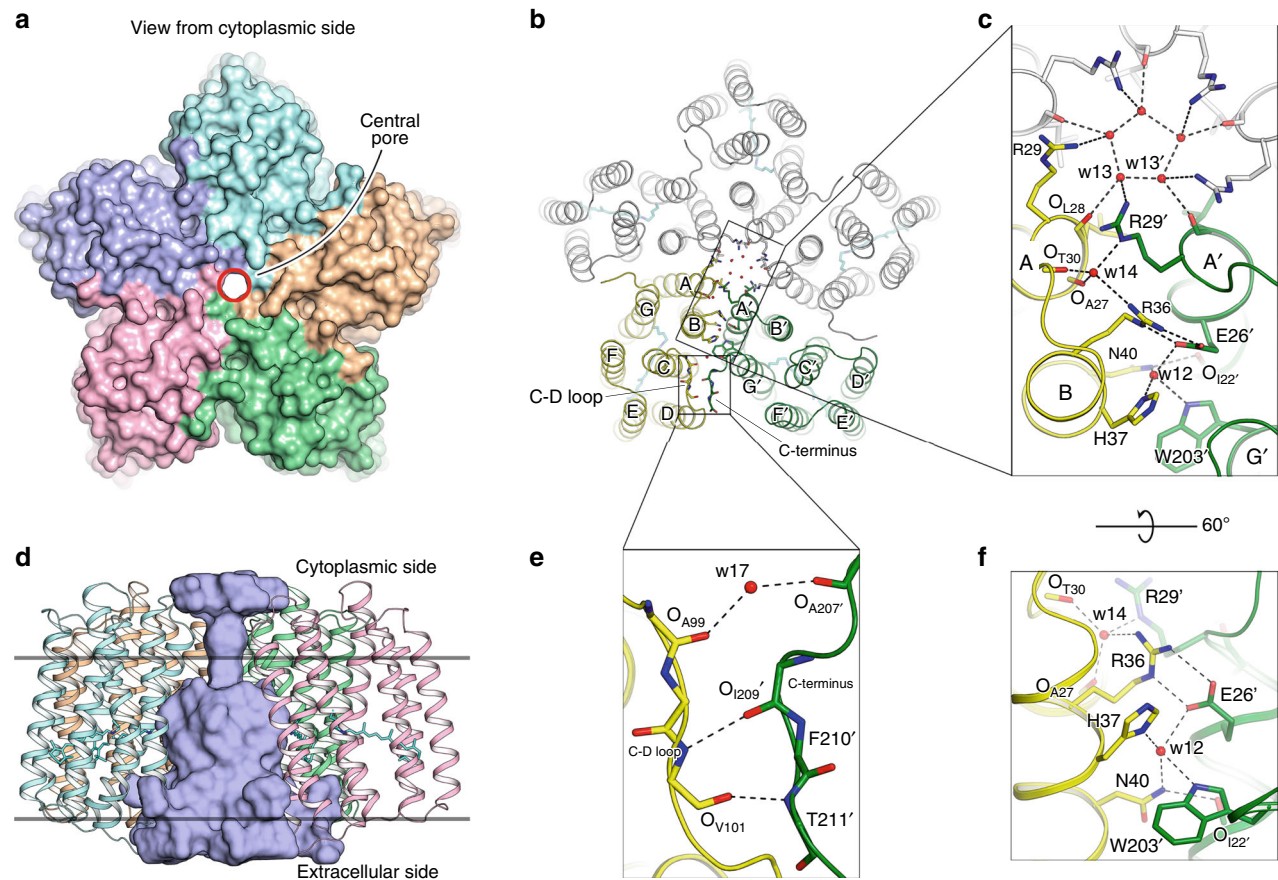

**Fig. 1** Architecture of OLPVRII pentamer and interprotomer contacts. **a** View from the cytoplasmic side. Surface representation of the pentamer. Central pore is contoured by a red circle. **b** View from the cytoplasmic side. Cartoon representation of the pentamer. Retinal cofactor is colored cyan. **c** Detailed view of the main region of interprotomer contacts. Protomers C, D and E are colored gray. **d** Side view of the pentamer. One protomer is hidden for clarity. Cavity inside the pentamer was calculated using HOLLOW[69] and is colored light blue, the hydrophobic membrane core boundaries are shown with solid horizontal lines. **e**, **f** Detailed view of the interprotomer contacts

during purification, and purified in detergent monomers, when placed into lipid environment, assemble into the pentamers again, particularly during crystallization. In addition, we expressed and purified a triple mutant E26A/R36A/W203A with three mutations of highly conserved amino acids that are located at the interface between the protomers and form hydrogen bonds between them. While the wild-type protein had the pentamer: monomer ratio being 4:1 in the elution profile, the triple mutant showed an approximately 1:1 ratio (Supplementary Fig. 6a). The fact that the triple mutant partially retained the pentameric assembly suggests that there are additional interactions between OLPVRII protomers, such as hydrogen bonding between C−D loop and C-terminus and hydrophobic cooperation. Taking into account that the key amino acids forming the pentamer are highly conserved in the viral rhodopsins of the second group (Supplementary Fig. 1), we suppose the pentameric assembly to be crucial for OLPVRII function.

The pentameric assembly was also observed for other microbial rhodopsins, such as PRs and NaRs[22] (Supplementary Fig. 7). Although the relative orientation of the protomers is similar for OLPVRII, KR2 and BPR HOT75 (Supplementary Fig. 7), the size and profile of the central pore together with the pentamerization contacts are completely different in these proteins (Supplementary Fig. 8). Oligomerization of PRs and NaRs affects the functionality of these proteins. For instance, in KR2, NaR from *Krokinobacter eikastus*, pentamerization is required for sodium pumping[23]. Thus, pentamerization could also be a key

determinant of the activity of group II viral rhodopsins, and a thorough analysis of the OLPVRII pentamer is required to elucidate its function.

**Central pore (channel) of the pentameric structure.** However, the contacts between the OLPVRII molecules described above are unusual for microbial rhodopsins. They lead to the formation of the bottle-shaped pore inside the pentamer formed by the A and B helices of the protomers (Figs. 1d, 2a). The narrowest section (vestibule) of the pore is formed by the Phe24, Leu28 and Arg29 of helix A and has the length of 11 Å and mean diameter of 6 Å. Notably, Phe24 and Arg29 are highly conserved in group II viral rhodopsins, while Leu28 is interchanged with other hydrophobic residues, Ile and Met (Supplementary Fig. 1). At the entrance of the pore, on the cytoplasmic side of the pentamer, the unique arrangement of Arg29 side chains allows the positioning of five neighboring positively charged amino acids. Hereafter, we denote this configuration Arg29 ring.

Deeper into the pore, oxygen atoms of Leu28 main chain coordinate five water molecules forming a pentagon stabilized by hydrogen bonds with the mean distance between two neighboring molecules of $2.6 \pm 0.2$ Å (mean ± s.e.m., for different pentamer molecules, Fig. 2b). This pentagon weakly interacts with Arg29-side chains as water molecules are located within $3.8 \pm 0.3$ Å (mean ± s.e.m., for different pentamer molecules) from the nearest arginine nitrogen atoms. The Leu28 and Phe24 side

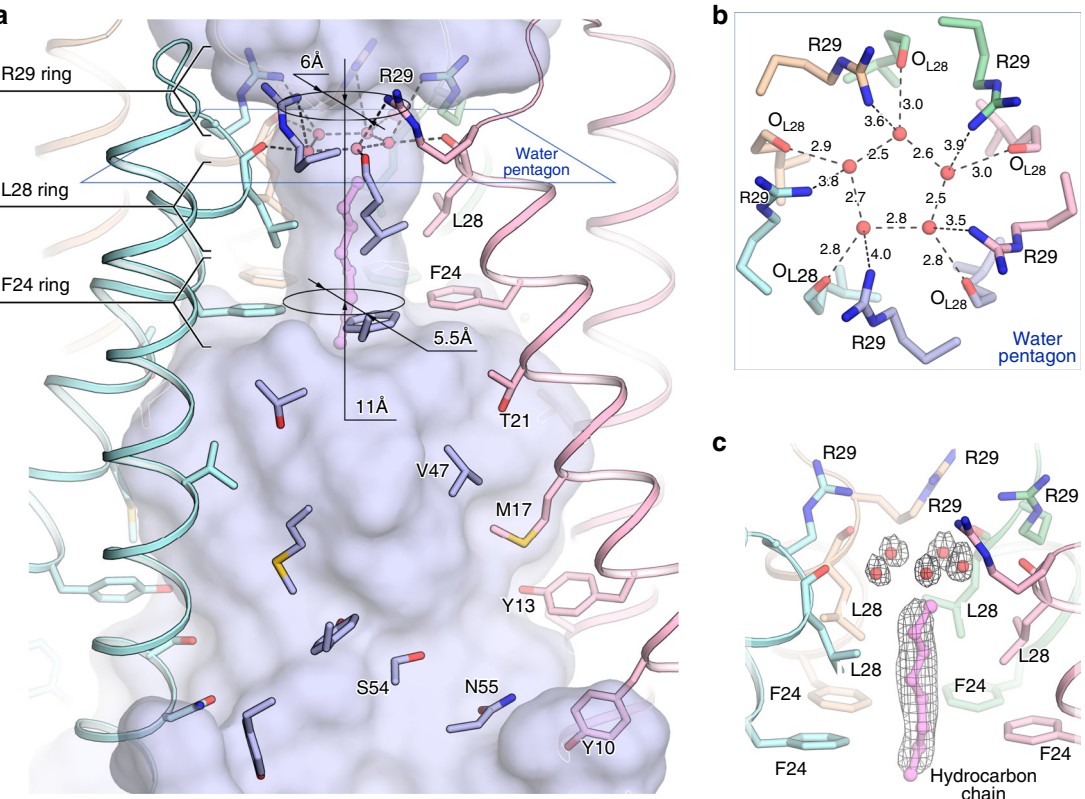

**Fig. 2** Central pore inside OLPVRII pentamer. **a** Overall central pore structure. One protomer is hidden for clarity. The lipid fragment is colored violet. **b** View from the cytoplasmic side on the water pentagon and interaction network between water molecules and the pore-lining OLPVRII residues. **c** Side view of the pore vestibule. Example of 2Fo-Fc electron density map is shown around the water pentagon and the hydrocarbon chain. The map is contoured at the level of 1.2σ. The hydrocarbon chain is colored violet

chains form an extremely hydrophobic part of the vestibule (Supplementary Fig. 9). Surprisingly, we observed a strong electron density along the pore axis in this region (Fig. 2c). We assigned it to a short hydrocarbon chain that plugs the pore. Phe24 side chain is oriented perpendicularly to the long axis of the pore forming the narrowest section of the vestibule with the diameter of only 5.5 Å. Closer to the extracellular side, Phe24 side chain forms the shoulder of the pore, where its diameter sharply increases to 26 Å. The wide extracellular part of the pore is lined with the residues Thr21, Val47, Met17, Ser54, Asn55, Tyr10 and Tyr13. The shoulder and the following part of the pore are paved with fragments of lipid molecules (Supplementary Fig. 9).

The comparison of the pores inside the oligomers formed by microbial rhodopsins with known structure shows the uniqueness of the channel in OLPVRII pentamer (Supplementary Fig. 8). Remarkably, however, a similar pore organization is observed in the transmembrane domains of pentameric Ligand-Gated Ion Channels (pLGICs) (Supplementary Fig. 10)[24]. Indeed, the ring of five negatively charged amino acids (Glu222 in each subunit) plays the role in the ion selectivity of a structurally characterized pLGIC from *Gloeobacter violaceus* (GLIC). Even more notably, the crystal structure of GLIC (PDB ID: 4HFI) contains similar pentagons of water molecules inside the transmembrane channel, presumably involved into hydration of the permeation ion, that are followed by the extremely hydrophobic region plugged with the carbon tails of detergent molecules[24]. The OLPVRII pore vestibule mimics the transmembrane channel of GLIC (Supplementary Fig. 10). Such an unusual narrow cytoplasmic part of the pore, its similarity to the transmembrane channels of pLGICs, and the evolutionary conservation of most residues involved in the formation of the pentamer and pore strongly suggest that the

central pore of OLPVRII functions as a channel. In pLGICs, opening and closure of the transmembrane channel are triggered by conformational changes starting with ligand binding at the active center of the extracellular domain. In contrast, structural rearrangements in microbial rhodopsins are initiated by absorption of light by the retinal cofactor that resides inside the protein. Therefore, OLPVRII is likely to function as a pentameric light-gated ion channel. To identify possible mechanisms of the central channel regulation in OLPVRII, we analyzed the protomer organization.

**Retinal pocket and extracellular part of OLPVRII.** In the OLPVRII pentamer, there are only slight differences between protomers. The root mean square deviation (RMSD) of the backbone atoms positions between different protomers varies from 0.17 to 0.20 Å. Thus, below, we describe only protomer A from the OLPVRII model.

Similarly to other microbial rhodopsins (Supplementary Fig. 11), each protomer of OLPVRII is organized as a bundle of seven transmembrane helices (A—G) connected by relatively short loops. The retinal cofactor is covalently bound to Lys195 through the Schiff base (RSB), and the electron densities indicate that the chromophore is in all-*trans* conformation (Supplementary Fig. 4c). OLPVRII consists of only 211 amino acids and is the smallest among microbial rhodopsins with known crystal structure. Compared to other rhodopsins, the length of the protein is reduced due to extremely short N- and C-termini as well as short loops that lack defined secondary structure. Because the differences in the structures of the OLPVRII and *Halobacterium salinarum* bacteriorhodopsin (bR) are likely to reflect

functional differences, we compare the structures of bR and OLPVRII in detail. The assembly of OLPVRII helices into a bundle differs from that in bR (PDB ID 1C3W), with RMSD of the backbone atoms positions of about 1.8 Å. Importantly, in OLPVRII, helix C is considerably longer, whereas helix F, conversely, is shorter than the corresponding helices of bR (Supplementary Fig. 12). The extracellular part of helix G is shifted inward for about 4 Å and the tilt and positions of other helices are also slightly altered in OLPVRII compared to bR (Supplementary Fig. 12).

The walls of the retinal-binding pocket are hydrophobic and similar to those of other rhodopsins except for the regions of the β-ionone ring and RSB. However, compared to bR, there are considerable differences in the adjacent amino acids: the highly conserved Pro186 is replaced with Gly173, Ser141 with Gly137 (like in PRs), and Thr142 with Phe138 (Fig. 3c and Supplementary Fig. 13). There are differences also at the cytoplasmic side of the RSB, where Leu93 of bR is replaced with Met83 and Ala215 with Ser194. Mutation A215T is the major change converting bR to a sensory rhodopsin, whereas Leu93 and homologous residues are functionally important and responsible, for example, for the color tuning in PRs[25,26]. These mutations lead to the shift of the RSB towards the extracellular part of the protein compared to bR, due to the steric conflict with Met83 and Ser194.

Like in bR, in OLPVRII, the RSB nitrogen is hydrogen bonded to a key water molecule Wat1 (Wat402 in bR) that donates hydrogen bonds to two anionic residues, Asp75 and Asp191 (Asp85 and Asp212 in bR, correspondingly). This arrangement stabilizes the positive charge of the RSB and is conserved in outward proton pumps and most other rhodopsins[27]. In OLPVRII, structural organization of the RSB that involves Asp75, Asp191 and Arg72 is similar to that of bR, but some differences are observed (Fig. 3a and Supplementary Fig. 13). The structure and mutational analysis described below suggest that Asp75 is the primary proton acceptor from RSB and the deprotonation of RSB in OLPVRII occurs in a manner similar to that of bR[28].

The space between Arg72 and the extracellular surface of the protein is notably different in OLPVRII compared to bR (Fig. 3 and Supplementary Fig. 14). Whereas in bR, Arg82 side chain is stabilized by hydrogen bonds with water molecules Wat403, Wat404, Wat405 and Wat407, and plays a key role in proton transfer[28], in OLPVRII, these water molecules are replaced by three Asn side chains, Asn69, Asn184 and Asn188. Importantly, Asn69 and Asn184 are highly conserved and Asn188 is completely conserved among the group II viral rhodopsins (Supplementary Fig. 1). While Arg82 was demonstrated to be a key element in the proton translocation mechanism in bR[28], and analogous arginine is found in most microbial rhodopsins, playing important roles in their functioning[29–31], we suggest that the strong stabilization of Arg72 in OLPVRII by three Asn side chains may affect its mobility and thus affect the function of the rhodopsin.

Strikingly different from bR, where the extracellular bulk is separated from the proton release group (Glu194–Glu204 pair), in OLPVRII, a large cavity protrudes from the extracellular protein surface down to Arg72 (Fig. 3), similarly to *Exiguobacterium sibiricum* rhodopsin (ESR) and ChR2 (Supplementary Fig. 15)[5,31]. This hydrophilic cavity is filled with water molecules and is formed by polar amino acids, including Arg183, Asp124,

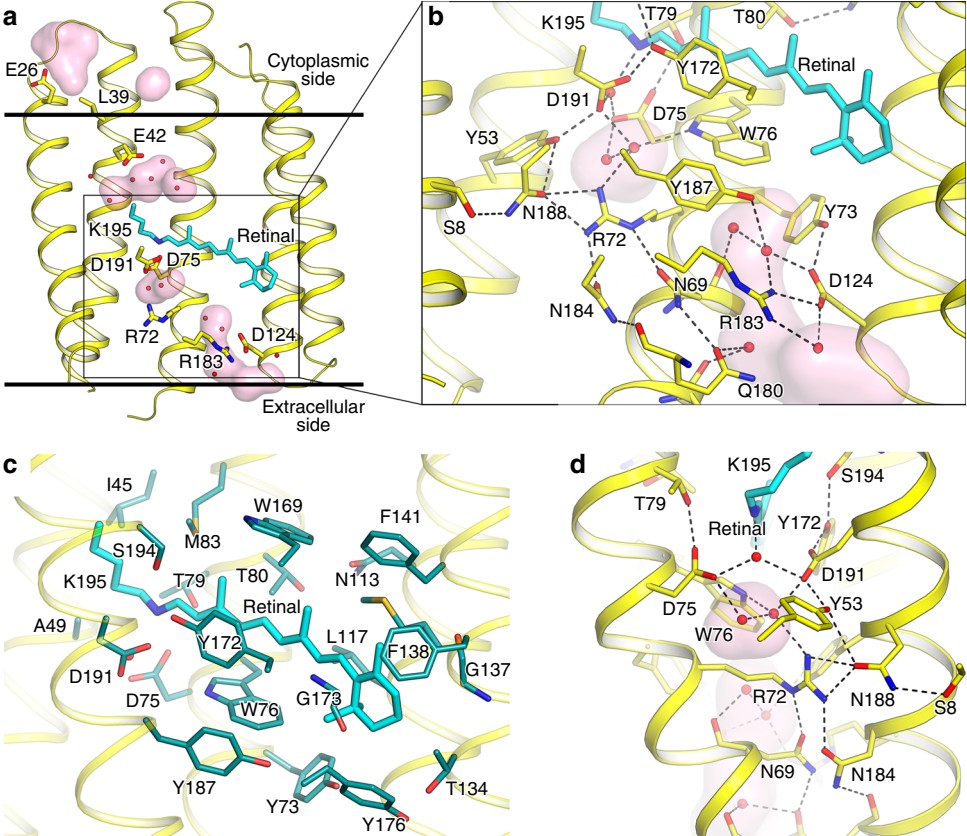

**Fig. 3** Structure of OLPVRII protomer, retinal-binding pocket and extracellular part. **a** Overall side view of protomer A. Helices F and G are hidden for clarity. Hydrophobic—hydrophilic boundaries of the membrane are shown with gray lines. **b** Detailed view of the extracellular part. **c** Detailed view of the retinal-binding pocket. Residues comprising the walls of the pocket are colored teal. **d** Detailed view of the RSB region. Helices A and B are hidden for clarity. Cavities inside the protein protomer are colored pink. Lys195 and covalently bound retinal are colored cyan

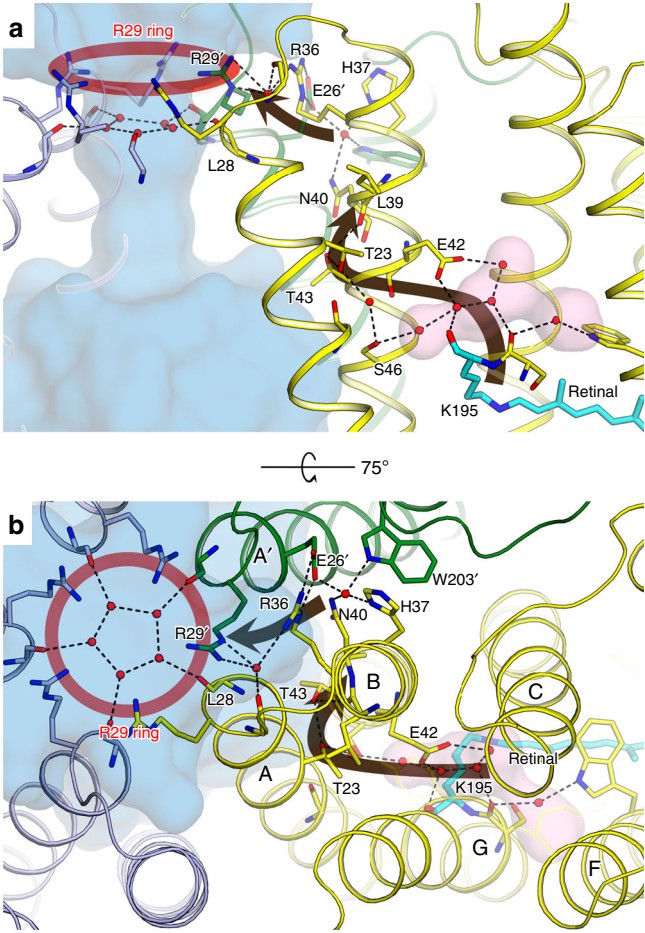

**Fig. 4** The cytoplasmic part of OLPVRII and its connection to the central pore. **a** Side view. One protomer is hidden for clarity. **b** View from the cytoplasmic side. The central pore is shown with the blue surface. The red circle indicates the R29 ring of the central pore. The brown arrows show the putative sequence of structural rearrangements transduced from RSB to the pore interface. Cavities inside the protein protomer are colored pink

Tyr73, Tyr187, Gln180 and Asn69 (Supplementary Fig. 14). These amino acids are densely hydrogen bonded and are also linked via a hydrogen bond network to Arg72, and further, to the RSB. Thus, in OLPVRII, Arg72 side chain separates the RSB region and the extracellular space. RSB deprotonation during the photocycle triggers structural rearrangements in the extracellular part of the protein and therefore can result in opening of the gate around Arg72 similarly to the way it might occur in ChR2[31].

**Cytoplasmic part of OLPVRII and its connection to vestibule.**
The cytoplasmic part of OLPVRII dramatically differs from those of all rhodopsins with known structures (Supplementary Fig. 16). First, there is no charged amino acid in the cytoplasmic half of the helix C (Val86 in OLPVRII at the position of Asp96 in bR) which would act as a proton donor to the RSB during the photocycle as in most of the outward proton-pumping rhodopsins[32]. In OLPVRII, the putative proton donor Glu42 is located in helix B (which corresponds to Thr46 in bR) and thus juxtaposed to the RSB. Glu42 is fully conserved in both groups of viral rhodopsins and site-directed mutagenesis results indicate that Glu42 indeed serves as a proton donor to RSB during the OLPVRII photocycle (see below).

The mechanism of Glu42 reprotonation remains unclear. Most likely, proton uptake proceeds from the side of Glu26, where the

cytoplasmic surface is concaved inside the protomer and reaches the side chain of Leu39, which is the only residue separating Glu42 from the bulk. In this case, Leu39 might play the role of a hydrophobic gate, similar to that of Leu93 in bR[33] (Supplementary Fig. 17). An alternative mechanism of Glu42 reprotonation could involve the uptake of the proton from the part of the protomer occupied by Asn94, Asn158, Ser157, Tyr201 and Tyr206, which together form a small hydrophilic cavity near the cytoplasmic surface of the protein.

Moreover, a large cavity is observed between the RSB and the Glu42 side chain. This cavity is hydrophilic and filled with five water molecules stabilized by hydrogen bonding with Glu42, Ser36 and main chain oxygens of Ser194 and Lys195 in helix G. A similarly positioned cavity exists in the wild-type ChR2 but, in this case, the cavity is hydrophobic, with no water molecules inside[31]. Hydrogen bonds protrude from Glu42 via water molecules Wat7, Wat8, Wat9, Wat10, Wat11 and Thr23 and Thr43 side chains to Leu39, which is located next to Asn40 from the pentamerization interface, strengthening interactions between helices A and B inside the protomer (Fig. 4). The distances between Glu42 and Wat9, as well as between Wat8 and Wat9, are extremely short, only 2.24 and 2.20 Å, respectively. The proton donor Glu42 as well as His37, Asn40 and Arg36 that form pentameric contacts in OLPVRII are all located within two turns of the same helix B. Therefore, in OLPVRII, the proton donor is rigidly connected to the oligomerization interface not only by the geometry and constraints of the α-helix but also by a continuous chain of strong hydrogen bonds through the water molecules in the hydrophilic cavity and polar residues of helix A. Furthermore, the dense hydrogen bonding network of the pentameric contacts protrudes directly to Arg29 ring (Fig. 4).

This distinct architecture of the cytoplasmic part, together with the conservation among group II viral rhodopsins of all amino acids responsible for the pentameric assembly of OLPVRII, strongly supports the idea that structural changes initiated by retinal isomerization upon absorption of light propagate directly from the RSB to the vestibule of the central pore. Indeed, the location of the proton donor Glu42 in the helix B, close to the RSB, its additional hydrogen bonding with helices A and G and the oligomerization interface, as well as the tight interprotomer contacts provide for an almost direct interaction between RSB and Arg29 that are located at the distance of 27 Å from each other (Fig. 4). The pore is formed by helices A and B, and accordingly, its shape and characteristics are determined by the hydrogen bonding network that involves and stabilizes these helices and depends on the retinal conformation (in particular, RSB orientation). Moreover, our attempt to trap an active state of the protein in the crystals led to a dramatic destruction of the crystals (like in the case of eye rhodopsin[25] and sensory rhodopsin[34]) evident by a drop of structural resolution from 2 to 20 Å. This fact indicates that large structural rearrangements occur in OLPVRII during the photocycle and gives another hint on the channel activity of the protein, where large structural rearrangements occur upon ion channel opening like in pLGICs[35].

Thus, detailed examination of the OLPVRII structure not only suggests that this rhodopsin is a pentameric light-gated ion channel, but also demonstrates the possible mechanisms of the channel functioning and regulation. To test these predictions, we carried out the functional assays of OLPVRII and molecular dynamics simulations.

**Spectroscopic characterization of OLPVRII.** The UV−Vis absorption spectra of OLPVRII solubilized in DDM show maximum absorbance of bound retinal at 514 nm. The position of the

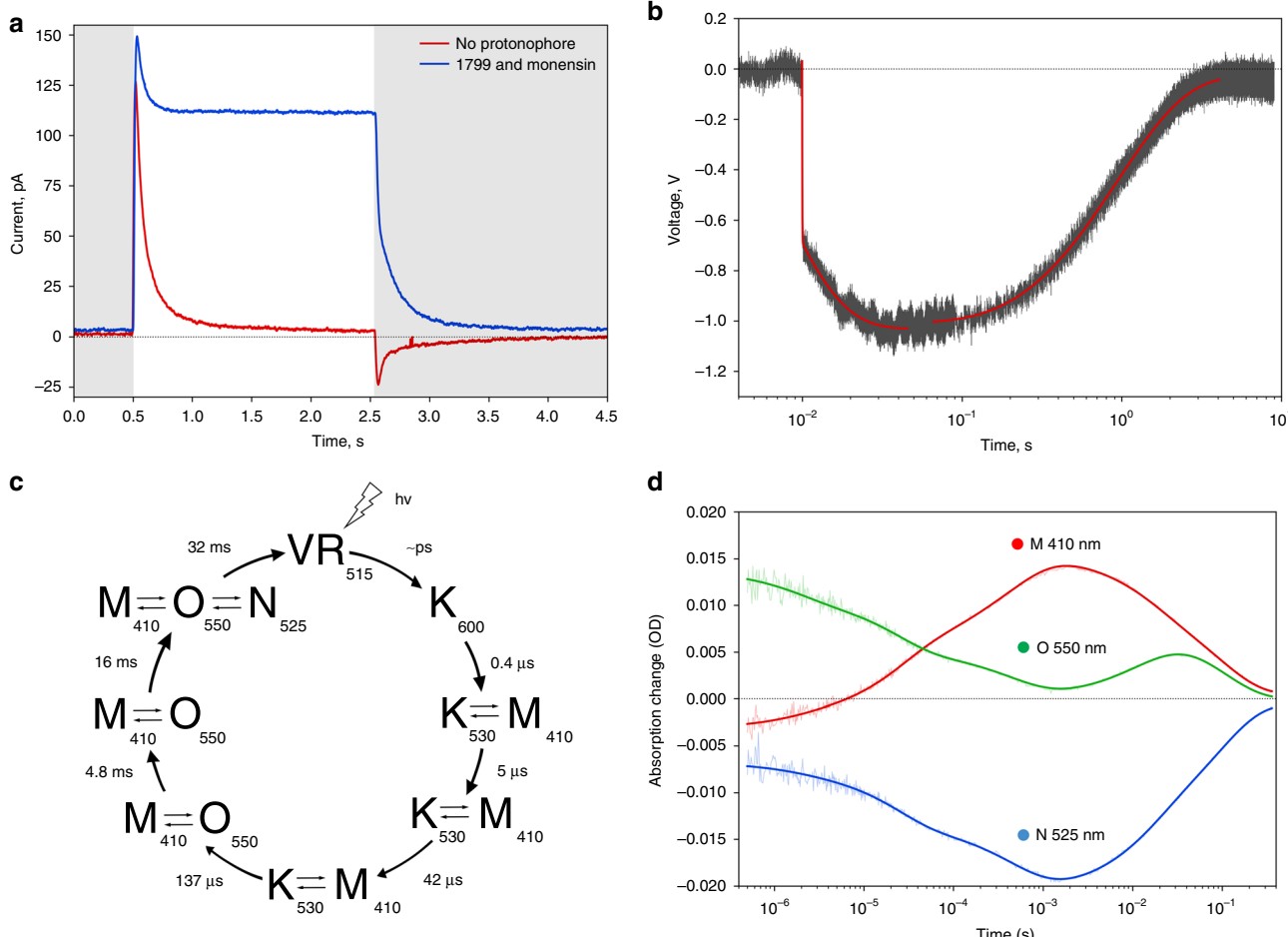

**Fig. 5** Functional characterization of OLPVRII. **a** BLM measurements of OLPVRII in *E. coli* polar lipid liposomes. Red curve shows the electrical signal on BLM under illumination without addition of protonophore, while blue curve represents the signal under the same conditions, but after addition of 1799 and monensin. Illumination onset is indicated by gray (light off) and white (light on) colors. **b** Time-resolved BLM experiment. Electrical signal (gray) was recorded after laser flash and fitted with the exponential decay curves (red). **c** Schematic representation of photocycle kinetics of the OLPVRII reconstituted into soybean liposomes. Lifetimes and absorption maxima of the intermediate states are indicated. **d** Time traces of the absorption changes of OLPVRII measured at 410, 525, and 550 nm. Source data are provided as a Source Data file

peak shifts depending on the buffer pH. We varied the pH of the OLPVRII sample in six-mix buffer from 7.58 to 1.93 and from 7.58 to 12.93 for two identical samples. At alkaline pH, the maximum absorbance of the retinal changed from 514 to 367 nm indicating the RSB deprotonation (Supplementary Fig. 18). A sigmoidal fit of the dependence of the absorbance at 514 nm on pH gave the pKa of the RSB of $10.36 \pm 0.08$ (point of half decay $\pm$ standard error). In contrast, at acidic pH, we observed a gradual shift of the retinal absorbance maximum from 514 to 548 nm indicating protonation of the RSB counterion Asp75. We fitted the dependence of the wavelength of the absorbance maximum on pH with a sigmoidal curve and obtained the pKa of the proton acceptor of $3.34 \pm 0.11$ (point of half decay $\pm$ standard error). The corresponding pKa values for bR are 12.2 and 2.7[36,37], whereas for PR, the pKa of the proton acceptor is 7.68[38]. These notable differences in pKa appear to stem, first, from the absence in viral rhodopsins of the histidine that is conserved in PRs and forms a hydrogen bond with the proton acceptor thus decreasing its pKa, and, second, the amino acid substitutions A215M and L93S, compared to bR, that alter the properties of the RSB.

The photocycle kinetics of monomeric OLPVRII expressed in the C41 strain and reconstituted into soybean lipid vesicles was measured at pH 7.5. Transient absorption changes after laser

illumination are shown in Fig. 5d at three characteristic wavelengths, 410, 530, and 550 nm. The dataset is best fitted by seven exponentials. The total length of the OLPVRII photocycle is about 70 ms, and the corresponding time constants were retrieved. The results of the global fit are shown in Fig. 5c.

The photocycle includes a microsecond part that is usually assigned to the release of the energized ion appearing as the RSB deprotonation, and a millisecond part of the ground state recovery by proton uptake. Like in most rhodopsins, the amount of the resolved kinetic intermediates is higher than the amount of spectrally distinct states which indicates the presence of spectrally silent transitions[39]. The representative absolute spectra of the intermediates are shown in Supplementary Fig. 19. In the microsecond part of the photocycle the RSB deprotonation occurs via a series of intermediates with characteristic times of $\tau_1 = 1\ \mu s$, $\tau_2 = 16\ \mu s$, $\tau_3 = 61\ \mu s$, and $\tau_4 = 340\ \mu s$. The first intermediate is composed of single K[540]-like spectral state, while P2−P4 are mixtures of K[540]- and M[410]-like states. The appearance of the blue-shifted M[410]-like intermediate with the deprotonated RSB in the range of hundreds of microsecond is typical for microbial rhodopsins. However, in contrast to bR[39] and many other rhodopsins, OLPVRII does not form an L-state, the same was obtained for proteorhodopsin[38]. Then, in the

millisecond time range, the $M^{410}$-like intermediate returns to the ground state ($\tau_5 = 10$ ms, $\tau_6 = 25$ ms, and $\tau_7 = 73$ ms), via a series of equilibrium states P5−P7 that consist of the mixture of $M^{410}$-, $N^{525}$-, and $O^{550}$-like states.

To validate the proposed mechanism of proton pumping, we analyzed E42Q and D75N mutants of solubilized OLPVRII, where the predicted negatively charged proton donor and acceptor, respectively, are replaced by polar residues. The mutant E42Q showed a photocycle closely similar to that of the wild-type protein, but with a considerably longer decay of the M-state, like in the case of the D96N mutant of bR[40], while the D75N failed to form the M-state (Supplementary Fig. 20). These experiments show that Asp75 is the proton acceptor of OLPVRII, as is the case in many other microbial rhodopsins, whereas Glu42 is the proton donor that is located in the helix B, in a sharp contrast to most known rhodopsins.

**Pumping activity of OLPVRII**. To access whether the viral rhodopsin is an ion pump, we conducted direct ΔpH measurements to evaluate the pumping activity of OLPVRII in *E. coli* cells and upon reconstitution into soybean lipid liposomes. In *E. coli*, no pH change of the external solvent under illumination was detected which could be attributed to the low yield of the protein. Then, we measured ΔpH with the purified protein reconstituted into lipid vesicles, where high protein density was reached. However, even in this system, and in contrast to the reference protein (bR expressed in *E. coli* as reported before[21]) no pH change of the external solution upon illumination was detected (Supplementary Fig. 21).

The electrogenic properties of OLPVRII were further explored in black lipid membrane (BLM) experiments that were designed as previously described[38]. Solubilized OLPVRII was reconstituted into *E. coli* polar lipids in the buffer 100 mM Tris-HEPES pH 7.4 (pH was chosen because this is close to the natural conditions in the Organic Lake[41]). Without an added protonophore, upon illumination of the proteoliposomes with continuous light, we observed characteristic photocurrents (Fig. 5a), where a fast upward deflection is followed by a subsequent slow decay back to zero current, whereas the stationary current is negligible. Adding ionophores 1799 and monensin[38] had no qualitative effect on the photocurrent, but a positive stationary current was observed indicating continuous pumping (Fig. 5a). The liposomes were prepared solely in the Tris-HEPES buffer. These ions are unlikely to be transported, so we conclude that the photocurrent is carried by protons.

To investigate which steps of the transport cycle are electrogenic, we performed time-resolved BLM measurements (Fig. 5b). Upon a laser flash the fast upward deflection has two time constants, $\tau_1' = 24$ μs and $\tau_2' = 6$ ms, whereas the slow decay of the current is characterized by two constants $\tau_3' = 90$ ms and $\tau_4' = 950$ ms. Comparing these time constants to the constants obtained in the flash photolysis kinetic measurements, we can assign the time constant $\tau_1'$ to the $K^{540} \rightarrow M^{410}$ transition, and the time constants $\tau_2'$ and $\tau_3'$ reflect the formation of late $O^{550}$- and $N^{525}$-like intermediates. The time constant $\tau_4'$ can be correlated with the unspecific discharge of the BLM[38].

Taking into account the arrangement of functionally important amino acids in the protein and the results of the BLM experiments, we conclude that, at least in liposomes, OLPVRII acts as an outward proton pump. However, the charge transfer per photocycle is small compared to that of the expected channel activity. Indeed, it was shown that ChR2 also possesses outward proton pumping activity, which is negligible compared to its ion channeling[42].

**Hydrophobic gate of the OLPVRII ion channel**. To test whether OLPVRII could function as a pentameric light-gated ion channel, we carried out unguided all-atom MD simulations of the protein embedded in a POPC lipid bilayer in a 200 mM NaCl aqueous solution (Supplementary Fig. 22). We observed spontaneous water flux into the central pore from both membrane sides except for the hydrophobic region occupied by the hydrocarbon chain (Fig. 6a). In the absence of these lipids, a continuous hydration pattern suggests a putative ion conduction pathway (Fig. 6a, b)[43]. However, closer inspection reveals a minimum in water density in the pore region lined by Phe24 and Leu28, where burst-like liquid −vapor oscillations occur on the nanosecond timescale. Calculation of pore radii shows a constriction in this region, with a minimum of 1.93 Å (Fig. 6c).

Arg29 creates a positive electrostatic potential at the cytoplasmic pore entrance that is separated by the hydrophobic constriction from the wide, partially negative, extracellular entrance (Fig. 6d and Supplementary Fig. 23). Free-energy profiles calculated for water, $Na^+$, and $Cl^-$ permeation show that the cytoplasmic side of the pore is highly anion selective, whereas the extracellular entrance is cation selective, with the highest barrier created by Phe24 and Leu28 (Fig. 6e). Reduction of the respective side chain volumes by alanine substitution increases the hydration of the constriction, lowers the energy barrier for ion permeation, and leads to an overall $Cl^-$-selective ion pathway in the double mutant. Along this pathway, we observed multiple spontaneous water and $Cl^-$ permeation events (Fig. 6f and Supplementary Fig. 23). Our simulations define OLPVRII as an anion channel, that is sterically open, but functionally closed by a hydrophobic gate formed by Phe24 and Leu28. Importantly, free-energy calculations demonstrate that charge reversal via the R29E mutation converts OLPVRII into a closed cation-selective channel, which is conductive in the F24A/L28A/R29E mutant, suggesting a similar gating mechanism as in the wild-type protein (Supplementary Fig. 23). Gating of OLPVRII thus resembles the hydrophobic gating mechanism in pLGICs and other biological ion channels[24,35,44–46]. Following light-induced isomerization of the retinal, activation of OLPVRII may propagate from the RSB via the hydrogen bond network toward the helices forming the pore. In Supplementary Fig. 24 we present a simple mechanistic model of OLPVRII anion channel activation. It is tempting to speculate that channel opening involves rotation or tilting of helix A, which moves Phe24 and Leu28 away from the pore and thereby lowers the effective hydrophobicity in the pore, causing hydrophobic gate wetting and permitting $Cl^-$ permeation. Indeed, similar rotation of the α-helix leads to significant displacement of hydrophobic residues (Phe80, Phe84 and Ile76) and the channel opening in pentameric bestrophin ligand-gated chloride channel BEST1[35] (Supplementary Fig. 25).

## Discussion

We show here that OLPVRII, a representative of the previously uncharacterized, growing family of rhodopsins encoded by large viruses, is a pentameric light-gated ion channel, most likely, specific for chloride, which is supported by MD simulations. The pentameric assembly of the protein was demonstrated in detergent, lipid membranes and crystals and is also supported by the high evolutionary conservation of the key amino acids that comprise the oligomerization interface. Although the overall structure of OLPVRII resembles those of other microbial rhodopsins, the viral rhodopsin shows some unique features, such as a distinct structure of the cytoplasmic part, a bottle-shaped central channel and an unusual position of the proton donor. There is unexpected similarity between the organization of the vestibule

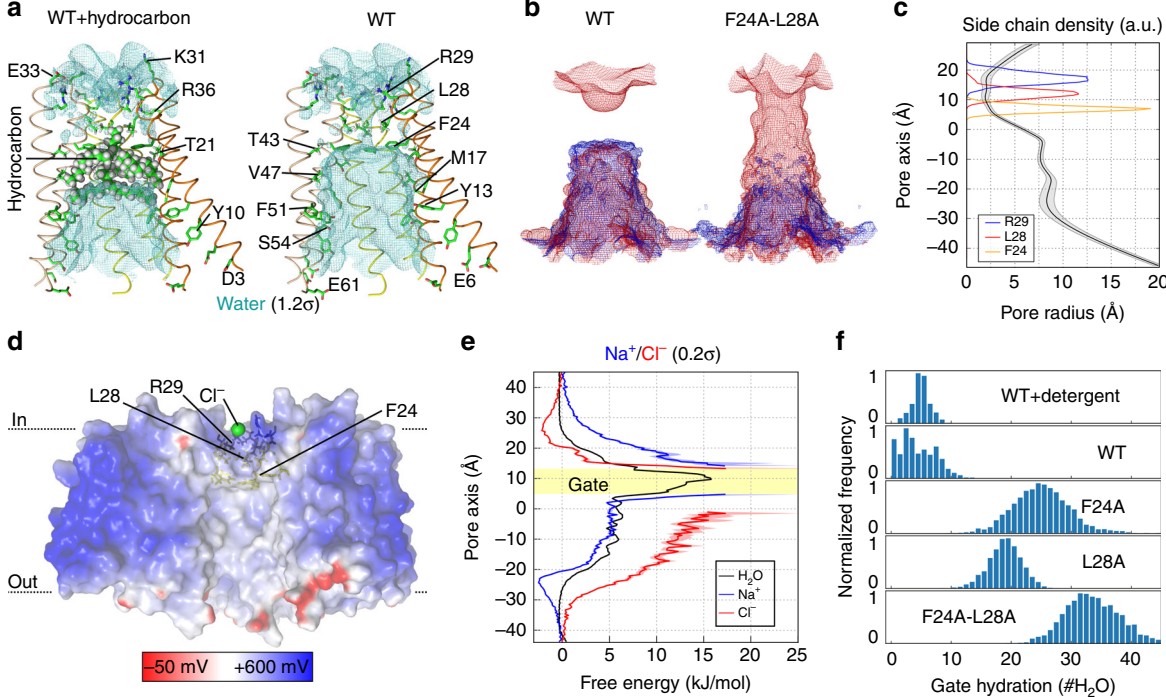

**Fig. 6** MD simulations reveal a closed anion permeation pathway. **a** Averaged water densities in absence or presence of the hydrocarbon chain along the central axis from unguided MD simulations contoured at 1.2$\sigma$ in side view. Helices A and B of three subunits are shown in cartoon representation and pore-lining residues as sticks. **b** Na$^+$ (blue) and Cl$^-$ (red) densities for WT and F24A-L28A mutant proteins in absence of the detergent contoured at 0.2$\sigma$. **c** Pore radii along the pore axis (positions relative to the protein center of mass) for the conformational ensemble sampled in WT simulations without hydrocarbon chain. Side chain positions of residues Phe24, Leu28, and Arg29 are shown. **d** Electrostatic potential distribution calculated from the MD simulations mapped onto the solvent-accessible surface of OLPVRII in side view. The two front subunits are hidden for clarity. **e** Potential of mean force profiles for water, Na$^+$, and Cl$^-$ permeation along the pore. **f** Histograms of the number of water molecules observed within the hydrophobic gate region for different mutations. Source data are provided as a Source Data file

in OLPVRII and unrelated pentameric ligand-gated ion channels. The role of rhodopsins in the virus infection remains to be established. Notably, some large viruses also encode other types of membrane channel proteins, such as small K+ channels[47,48], suggesting that modulation of ion homeostasis in infected cell could be important for virus reproduction. One plausible possibility is that viral rhodopsins affect ion currents across the membrane of an infected protist hosts in response to light and accordingly influence their behavior, such as phototaxis[16,17].

## Methods
**Protein sequence alignment and phylogenetic analysis.** The dataset of group II viral rhodopsins was constructed by searching the NCBI nonredundant protein sequence databases along with the TARA[49] metagenomic sequences using BLASTP and TBLASTN. An alignment of 213 identified complete or partial sequences of group II viral rhodopsins including five previously reported was constructed using MUSCLE[50] with default parameters. The conservation of the functionally important amino acids was investigated using the full sequence alignment (Supplementary Data 1). For the sake of clarity, for the Supplementary Fig. 1 we used a reduced number of sequences. To obtain a representative set of 25 TARA metagenomic sequences, we used the CD-HIT suite[51] with default parameters and 60% identity cutoff. Then we combined 25 representative sequences with five sequences reported before and aligned them using MUSCLE at default parameters.

Phylogenetic reconstruction for the Supplementary Fig. 2 was performed using PHYLIP Neighbor Joining method as implemented in the UGENE software[52] with the following parameters: Jones−Taylor−Thornton model, transition/transversion ratio = 2.0, no gamma distribution applied. The unrooted phylogenetic tree was visualized using iTOL[53]. The full list of rhodopsin sequences that were used for sequence alignment and phylogenetic tree construction is given in the Supplementary information.

For Supplementary Fig. 3, reference rhodopsin superfamily proteins were collected from refs. [16,54]. Sequence names contain Genbank accessions after "@" symbol. GenBank Identification (GI) numbers are shown in parentheses for metagenomic sequences used in ref. [16]. Protein sequences were aligned using the MUSCLE with default parameters[50], columns containing a large fraction of gaps

(greater than 30%) and nonhomogenous columns defined as described previously[55] were removed from the alignment. The tree was reconstructed using PhyML[56] with LG substitution model, gamma-distributed site rates, empirical amino acid frequencies and a Bayes branch support values. The root of the tree was enforced between heliorhodopsins and the rest of the microbial rhodopsins[54]. The tree in Newick format and the multiple protein alignment used for the reconstruction can be found at ftp://ftp.ncbi.nih.gov/pub/yutinn/viral_rhod_2019/.

**Cloning.** We optimized the nucleotide sequence of OLPVRII (UniprotID F2Y2Z0) for *E. coli* expression using the GeneOptimizer™ software (Life Technologies, USA) and synthesized it commercially (Eurofins). We introduced the gene into the pSCodon 2.1 expression vector (Delphi Genetics) via NdeI and XhoI restriction sites and appended at the 3′ terminus additional GSGIEGRSGAPHHHHHHHH* tag, which was used for metal-affinity chromatography purification and contains FXa cleavage site. The same gene was introduced into pEKT expression vector, pET vector derivative (Novagen), via XbaI and XhoI restriction sites.

***E. coli* expression and purification of the protein.** We expressed the protein as described previously[14,21] with slight modifications. We transformed *E. coli* SE1 (Delphi Genetics) or C41(DE3) (Lucigen) cells with the expression plasmid and grew them at 37 °C in 2 l shaking at 120 rpm baffled flasks in an autoinducing medium ZYP-5052 [57] containing the required antibiotic. At optical density OD$_{600}$ of 1.0 we decreased the temperature to 20 °C and supplemented the media with 10 μM all-*trans*-retinal. After 14 h cultivation the cells were disrupted in M-110P Lab Homogenizer (Microfluidics, USA). Then we isolated total membranes by ultracentrifugation, resuspended them in 20 mM Tris-HCl pH 8.0, 100 mM NaCl and solubilized overnight in DDM (*n*-Dodecyl-β-D-Maltopyranoside, Anagrade, Anatrace, USA or crystallography grade, Cube-Biotech, Germany). The insoluble fraction was removed by ultracentrifugation (90,000 × *g*, 1 h, 4 °C). The supernatant was loaded on Ni-NTA column (Qiagen, Germany) and after washing the column with five volumes of 50 mM NaH$_2$PO$_4$/Na$_2$HPO$_4$ pH 8.0, 100 mM NaCl, 50 mM imidazole, 0.3% DDM buffer we eluted the protein in a buffer containing 50 mM NaH$_2$PO$_4$/Na$_2$HPO$_4$ pH 8.0, 100 mM NaCl, 0.5 M imidazole and 0.3% DDM. Eluted protein was concentrated using microfiltration (MW cutoff 30 kDa, Amicon Ultra, Millipore, Germany). Then we applied it to 30 ml Superdex 200i (GE Healthcare, Germany) column equilibrated with 50 mM NaH$_2$PO$_4$/Na$_2$HPO$_4$ pH 8.0, 100 mM NaCl, 0.2% DDM, and pooled a peak of colored functional protein.

The total yield of the functional purified protein was approximately 2−3 mg per 5 l of culture. After size exclusion chromatography, the protein was divided into two fractions, monomers and pentamers, according to the protein size appearing in the elution profile (Supplementary Fig. 6). Pentamerization was confirmed by cross-linking experiments with the solubilized protein. The triple mutant E26A/R36A/W203A was purified on different column (the same type, but with slightly different volume) that could be a reason of the monomer and pentamer peaks shift (Supplementary Fig. 6). For crystallization and functional tests, we used the monomeric fraction unless otherwise indicated. Finally, using microfiltration we concentrated homogeneous protein to 20 mg/ml, froze it in liquid nitrogen and stored it at −80 °C until used for crystallization or functional measurements. Typically, we obtained the monomeric fraction of the protein with the peak ratio $A_{280}/A_{520}$ of about 1.0 −1.3 and pentameric fraction with the peak ratio of about 1.3−1.7. Generally, peak ratio of the pentameric fraction is slightly higher probably due to aggregate impurities. Low peak ratio indicates a very high purity of the samples. The purity of the samples is also illustrated with SDS-PAGE (Supplementary Fig. 6B, lanes 1 and 2). SE1 strain showed an instability with OLPVRII-bearing pSC plasmid and its production was discontinued; therefore, we used C41 strain bearing OLPVRII gene in pEKT plasmid. Protein from C41 did not show considerable differences to SE1-expressed protein; therefore, we used it in the later stages of the study. Mutant proteins were expressed and purified similarly.

**Reconstitution of the protein into liposomes and nanodiscs**. We dissolved 20 mg of phospholipids (asolectin from soybean, Sigma-Aldrich) in $CHCl_3$ (Chloroform ultrapure, Applichem Panreac) and dried them under a stream of $N_2$ in a glass vial. Residual solvent was removed by overnight incubation under vacuum. Then we resuspended the dried lipids in 1 ml of 0.15 M NaCl buffer supplemented with 2% (w/v) sodium cholate, clarified the mixture by sonication and added OLPVRII at a protein/lipid ratio of 5:100 (w/w). The detergent was removed by incubation with detergent-absorbing beads (Amberlite XAD 2, Supelco). To ensure the ample removal of the detergent, we added the beads to the suspension 3−4 times. The volume of the beads was about one-third of the total volume of liposome mixture. We incubated the suspension with the beads by at least 2 h stirring. The last incubation was prolonged to 16 h. Then, the mixture was dialyzed against 0.15 M NaCl (adjusted to a desired pH) at 4 °C for 1 day with several buffer changes to obtain the desired pH. The described protocol leads to the formation of unilamellar vesicles with the size of 100−300 nm depending on the lipid composition that was confirmed by dynamic light scattering. We used obtained liposomes for the measurement of pump activity with pH electrode.

We prepared liposomes for BLM measurements with the following modifications: (a) E. coli polar lipids (Anatrace, Affymetrix, USA) were used instead of asolectin as a host lipid for liposomes; (b) to achieve $Na^+$- and $Cl^-$-free environment lipids were resuspended in 100 mM Tris-HEPES pH 7.4 buffer supplied with 2% (w/v) cholic acid in the same buffer and liposomes were not dialyzed against NaCl; (c) to improve the signal-to-noise ratio lipid-to-protein ratio was increased to 20:100.

The OLPVRII-containing nanodiscs were assembled using the following protocol[58]. 1,2-dimyristoyl-sn-glycero-3-phosphocholine (DMPC, Avanti Polar Lipids, USA) and an elongated in-house-produced MSP1E3 version of apolipoprotein-1 was used as a lipid and scaffold protein, respectively; the molar ratio during the assembly was DMPC:MSP1E3:OLPVRII = 100:2:3. The empty nanodiscs and aggregates were removed by size exclusion chromatography.

For crosslinking experiments liposomes from E. coli polar lipids turned out to be not suitable, as a natural mixture of lipids contains considerable amount of $NH_2$-groups that block the crosslinking reaction. Therefore we used an artificial mixture of POPC:POPS lipids (1-palmitoyl-2-oleoyl-glycero-3-phosphocholine and 1-palmitoyl-2-oleoyl-sn-glycero-3-phospho-L-serine from Avanti Polar Lipids, USA) in 4:1 weight ratio. The liposomes were prepared as described above.

**Crosslinking**. We performed crosslinking according to the following protocol[59] with the following modifications. We placed glutaraldehyde solution directly to the bottom of the well of 24-well Falcon Tissue Culture Plate (Thermo Fisher Scientific, Germany). OLPVRII was loaded on the microbridges, then the setup was covered by cover slips and incubated at 4 °C. We varied the concentration of the protein and glutaraldehyde to find the optimal condition for incomplete crosslinking of the proteins. This allowed us to observe in the denaturing conditions of SDS-PAGE the array of artificial multimers formed by several crosslinked protomers of the natural multimer. Using the molecular weight of the highest visible multimer and the number of the appeared multimers, we could make a conclusion about an aggregate state of the protein in a sample. We used for the experiments the following samples: solubilized in DDM multimeric fraction of OLPVRII, reconstituted into E. coli polar lipids and POPC/POPG mixture (ratio of 4:1 by weight) monomeric fraction of OLPVRII. The optimal conditions for solubilized protein were 2 mg/ml protein, 10% glutaraldehyde, 8 h incubation (Supplementary Fig. 6B, lane 4). For the monomeric protein reconstituted into POPC/POPG mixture, we incubated the sample for 8 h, the concentration of the protein in liposomes was 1 mg/ml and glutaraldehyde—10% (Supplementary Fig. 6B, lane 5). For the E. coli polar lipids, probably due to the presence of $NH_2$-groups in the lipid mixture, the crosslinking was not so efficient, so the formation of the pentamers was not observed.

**Measurements of pumping activity in E. coli and liposomes**. We transformed E. coli cells of strain C41(DE3) with the pSC expression plasmid and grew at 37 °C in shaking baffled flasks in an autoinducing medium, ZYP-5052 containing 100 mg/l ampicillin. Expression of OLPVRII was induced at optical density $OD_{600}$ of 0.6–0.7 with 1 mM isopropyl $\beta$-D-1-thiogalactopyranoside (IPTG) and the media was supplemented with 10 μM all-trans-retinal. Three hours after induction, we collected the cells by centrifugation at $3000 \times g$ for 10 min and then washed three times with an unbuffered salt solution (100 mM NaCl and 10 mM $MgCl_2$) with 30-min intervals between the washes to allow exchange of the ions inside the cells with the bulk. After that the cells were resuspended in 100 mM NaCl solution and adjusted to an $OD_{600}$ of 8.5. The measurements were performed in 3 ml aliquots of stirred cell suspension kept at 0 °C. Cells were illuminated for 5 min with a halogen lamp (Intralux 5000-1, VOLPI) and the light-induced pH changes were monitored with a pH meter (LAB 850, Schott Instruments). Similar experiments were performed on proteoliposome suspension, which was illuminated for 15 min with a halogen lamp (Intralux 5000-1, VOLPI) at 0 °C and then kept in the dark for another 15 min. As a positive control we used proteoliposomes with HsbR that was expressed in E. coli and purified similarly to OLPVRII with the major difference that purification steps starting from Ni-NTA resin wash were conducted in buffer with pH 6.0. The detailed protocol is described in our previous work[21].

**Spectroscopic characterization of OLPVRII**. We recorded the absorption spectra using the Shimadzu UV-2401PC spectrophotometer. Fitting at lower pH values of the dependence of the retinal absorbance maximum on the pH value with Boltzmann sigmoidal curve gave the pKa of the proton acceptor (point of half decay ± standard error). Fitting at higher pH values of dependence of the absorbance at 514 nm on the pH value with Boltzmann sigmoidal curve gave the pKa of the Schiff base (point of half decay ± standard error). To access the photocycle kinetics of viral rhodopsin, we used two different laser flash photolysis setups. The first setup was similar to that described by Chizhov and co-workers[39]. Surelite II-10 Nd:YAG laser (Continuum Inc, USA) was used providing pulses of 5 ns duration at 532 nm wavelength and energy of 3 mJ/pulse. We placed the samples between two collimated mechanically coupled monochromators (1/8 m model 77250, Oriel Corp., USA), and the probing light (Xe-arc lamp, 75 W, Osram, Germany) passed the first monochromator sample and arrived after a second monochromator at a PMT detector (R3896, Hamamatsu, Japan). Two digital oscilloscopes (LeCroy 9361 and 9400A) were used to record the traces of transient transmission changes in two overlapping time windows.

The second setup was as follows. Brilliant B Nd:YAG laser (Quantel, France) was used providing pulses of 4 ns duration at 500 nm wavelength and energy near 2 mJ/pulse. Samples were placed between two collimated mechanically coupled monochromators (LSH-150, LOT, Germany). The probing light (Xe-arc lamp, 75 W, Hamamatsu, Japan) passed the first monochromator sample and arrived after a second monochromator at a PMT detector (R12829, Hamamatsu, Japan). Two digital oscilloscopes (Keysight DSO-X 4022A) were used to record the traces of transient transmission changes in two overlapping time windows.

On both setups we recorded transient absorption changes starting from 700 ns after the laser pulse until completion of the photocycle; at each wavelength from 330 to 730 nm in 10-nm steps, 25 laser pulses were averaged to improve the signal-to-noise ratio. We analyzed the data sets using the global multiexponential nonlinear least-squares fitting program MEXFIT as was reported earlier[39].

We measured the photocycle kinetics of OLPVRII at pH 7.5 and temperatures between 0 and 30 °C with 10° intervals. For the experiments solubilized monomeric protein expressed in C41 strain was reconstituted into soybean lipid vesicles. Along with this experiment we conducted additional tests. For the SE1-expressed protein we measured the photocycle of both monomers and pentamers for the protein in three different environments: solubilized in DDM, reconstituted into soybean lipids and nanodiscs, and did not observe any significant photocycle differences. The photocycle of the monomers in detergent was shorter (35 ms comparing to 70−100 ms in the other environments), but the order and appearance of the intermediates were the same for all samples. The fact that photocycles measured for a membrane protein in different environments are very similar is not unique to OLPVRII. Whereas the replacement of the native lipid matrix by detergents significantly changes the kinetics of bR, in opposite, the photocycles of pHR and pSRII were found almost identical upon solubilization in DM[60,61]. Moreover, as we observed the deprotonation of the Schiff base at higher pH values, to ensure the robustness of the data we additionally measured the photocycles of the detergent-solubilized OLPVRII monomers at pH 8.0, 7.0, and 6.0. The protein demonstrated identical behavior at these pH values.

**BLM measurements**. The BLM setup was similar to that described by Bamberg and co-workers[38,62]. Optically BLMs with an area of $\sim 10^{-2} cm^2$ were formed across a hole between the two compartments of a cuvette filled with an electrolyte solution (100 mM Tris-HEPES, initial pH 7.4). The membrane-forming solution consists of n-decane with addition of 1.5% (w/v) diphytanoyl-phosphatidylcholine (Avanti Biochemicals, Birmingham, AL) and 0.025% (w/v) octadecylamine (Riedel-de-Haën, Hannover, Germany). After the addition of OLPVRII-containing pro-teoliposomes, photosensitivity of the samples reached maximal current amplitudes after ∼90 min. We carried out the experiments under illumination of either mercury arc lamp (Osram HBO 103W) or excimer laser-pumped dye laser

(LPX105MC, Lambda Physics, Göttingen, Germany). Afterwards, a combination of ionophores (2–3 µM 1799 ((2,6-dihydroxy)-1,1,1,7,7,7-hexafluoro-2,6-bis(tri-fluoromethyl)heptane-4-one) and 5 µM monensin) was added effectively permeabilizing the membrane system (final conductance 50–100 nS). We measured the photocurrents under short-circuit conditions, so that no external driving force is generated. All BLM experiments were carried out at room temperature. Signal recording and shutter/laser-triggering was carried out by Pclamp8 software via a Digidata 1200 Interface (Axon Instruments).

**Crystallization and structure determination**. The crystals were grown using the *in meso* approach[63], similar to our previous studies[10,21]. We mixed the solubilized protein with premelted monoolein at 42 °C (Nu-Chek Prep) to form a lipidic mesophase. Final protein concentration in the phase was 20 mg/ml. Then we spotted 150 nl drops of mesophase on a 96-well LCP glass sandwich set (Paul Marienfeld GmbH, Germany; bottom slide is covered with a 0.2 mm high spacer that has 96 recesses of 5 mm diameter) and overlaid them with 400 nl of precipitant solution using NT8 crystallization robot (Formulatrix). The crystals grew at 22 °C temperature and appeared in 6−12 weeks. The best crystals were obtained with a protein concentration of 25 mg/ml and 0.2 M sodium malonate, pH 4.6, 15% PEG 550 MME. All crystals were harvested using micromounts (MiTeGen, USA) and were flash-cooled and stored in liquid nitrogen for further crystallographic analysis.

X-ray diffraction data were collected at P14 beamline of the PETRAIII, Hamburg at 100 K, with an EIGER 16M detector. We processed diffraction images with XDS[64] and scaled the reflection intensities with AIMLESS from the CCP4 suite[65]. The crystallographic data statistics are presented in Supplementary Table 1. Reference model (archaerhodopsin-2, PDB 2EI4) for molecular replacement was chosen with the MoRDa pipeline. Initial phases were successfully obtained in P2₁ space group by an Automated Model Building and Rebuilding using Autobuild. The initial model was iteratively refined using REFMAC5[66], PHENIX[67] and Coot[68]. The cavities inside the protein were calculated using HOLLOW[69]. For the calculation of the central pore cavity, we used the "surface probe" value of the "hollow.txt" parameters file of 25 Å. Hydrophobic−hydrophilic boundaries of the membrane were calculated using PPM server[70].

**Molecular dynamics simulations**. We carried out all-atom MD simulations of OLPVRII using the CHARMM36m force field with the TIP3P water model[71]. The simulation box contained an equilibrated 1-palmitoyl-2-oleoyl-sn-glycero-3-phosphocholine (POPC) lipid bilayer, surrounded by 200 mM NaCl aqueous solution. OLPVRII was modeled using the residue range 1–211 and embedded into the bilayer using g_membed[72]. We assigned Glu42 a protonated state, all other amino acids were modeled in their default ionization state to reflect the most probable state at neutral pH based on pKa calculations using PROPKA 3.1[73,74]. Hydrocarbon chains bound to the central pore were modeled as one octane and five dodecane molecules.

We performed simulations using GROMACS 2018[75] in the NPT ensemble with periodic boundary conditions and an integration time step of 2 fs. Temperature was maintained at 310 K using the velocity-rescaling thermostat; pressure was kept at 1 bar using the semi-isotropic Parrinello−Rahman barostat as described recently[76]. Lennard−Jones interactions were truncated at 12 Å with a force switch smoothing function from 10 to 12 Å. Electrostatic interactions were calculated using particle mesh Ewald method and a real space cutoff of 12 Å. Simulation systems were equilibrated with position restraints on the protein heavy atoms for 400 ns, followed by ~20 ns with backbone-only position restraints. Subsequently, we performed 2–3 independent production runs of 2 µs length for OLPVRII with and without bound hydrocarbon chains in the pore, and F24A, L28A, R29E, F24/L28A, and F24/L28A/R29E mutants without the chains (Supplementary Fig. 22).

For consistency, all analyses of the simulations are based on the time windows from 1400 to 1800 ns in each simulation. We analyzed water and ion density distributions in the MD trajectories using GROmaps[77] and MDAnalysis[78] and calculated electrostatic potential distributions from the charge densities in the MD simulations using the Poisson equation as implemented in PMEPot[79]. Pore radii were calculated using HOLE[80] for an ensemble of conformations observed in the MD trajectories. Water, Na⁺, and Cl⁻ distributions in the free MD simulations were analyzed to calculate potential of mean force profiles along the pore axis via $G(z) = -RT \ln \frac{n(z)}{n_{bulk}}$, where $G$ is the free energy, $T$ the temperature, and $R$ the gas constant. Densities were analyzed within a cylinder placed along the pore axis with a radius of 36.23 Å (resulting in a cross-sectional area identical to the area of OLPVRII in the membrane plane); $n(z)$ is the density within a membrane slice at axial position $z$, and $n_{bulk}$ is the density in the bulk phase. We calculated standard deviations of energies and pore radii using block bootstrap sampling.

**Reporting summary**. Further information on research design is available in the Nature Research Reporting Summary linked to this article.

## Data availability

Data supporting the findings of this manuscript are available from the corresponding author upon reasonable request. A reporting summary for this Article is available as a Supplementary Information file. The source data underlying Figs. 5a, b, d, 6c, e, f and Supplementary Figs. 6a, b, 18a−f, 19, 20, 21, 22a, b, 23b are provided as a Source Data file. Coordinates and structure factors have been deposited in the Protein Data Bank (PDB) under the accession code 6SQG.

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

## Acknowledgements

We are grateful to Dr. Igor Melnikov for obtaining the original molecular replacement solution. We acknowledge the Structural Biology Group of the European Synchrotron Radiation Facility (ESRF) and The European Molecular Biology Laboratory (EMBL) unit in Hamburg at Deutsche Elektronen-Synchrotron (DESY) for granting access to the synchrotron beamlines. This work was supported by the common program of Agence Nationale de la Recherche (ANR), France and Deutsche Forschungsgemeinschaft (DFG), Germany (ANR-15-CE11-0029-02/FA 301/11-1), by the DFG Research Unit FOR 2518 (DynIon, project P4 to JPM, MA 7525/1-1), and by funding from Frankfurt: Cluster of Excellence Frankfurt Macromolecular Complexes by the Max Planck Society (to E.B.) and by the Commissariat à l'Energie Atomique et aux Energies Alternatives (Institut de Biologie Structurale)–Helmholtz- Gemeinschaft Deutscher Forschungszentren (Forschungszentrum Jülich) Special Terms and Conditions 5.1 specific agreement. This work used the platforms of the Grenoble Instruct-ERIC center (ISBG; UMS 3518 CNRS-CEA-UJF-EMBL) within the Grenoble Partnership for Structural Biology (PSB). Platform access was supported by FRISBI (ANR-10-INBS-05-02) and GRAL, a project of the University Grenoble Alpes graduate school (Ecoles Universitaires de Recherche) CBH-EUR-GS (ANR-17-EURE-0003). Spectroscopic characterization, time-resolved absorption spectroscopy, and crosslinking experiments were supported by RSF-DFG grant (Helmholtz—RSF Joint Research Groups grant (RSF No. 19-44-06302)). D.B. was supported by grant ANR-14-CE09-0028. F.R.-V. was supported by grant VIREVO CGL2016-76273-P (AEI/FEDER, EU) (co-founded with FEDER funds). D.J.M. was funded by the Deutsche Forschungsgemeinschaft under Germany's Excellence Strategy—EXC 2155 "RESIST"—Project ID 39087428. N.Y. and E.K. are funded through the Intramural Research Program of the National Institutes of Health of the USA. V.B. is thankful to the Ministry of Science and Higher Education of the Russian Federation (project № 6.9909.2017/6.7) for personal support. K.K., A.A., I.G., V.B., D.Z. were supported by grant 17-00-00164komfi (Russian Foundation for Basic Research). D.Z. and R.A. were supported by grant 6.3157.2017 (the Ministry of Science and High

Education of the Russian Federation). The authors gratefully acknowledge the computing time granted by the JARA-HPC Vergabegremium and VSR commission on the super-computer JURECA at Forschungszentrum Jülich.

## Author contributions

D.B. made cloning; D.B. and C.B. expressed and purified the proteins; D.B., I.C., D.S., D.V. measured the photocycle kinetics and analyzed the corresponding data; D.B., T.M., T.R. and Y.A. performed the BLM measurements; D.B., D.Z., A.A. and V.S. performed measurements of pumping activity and crosslinking experiments; D.B., R.A. crystallized the protein with the help of K.K.; V.P. took part in the initial crystallization trials and supervised them; V.P. and K.K. collected the diffraction data; A.P., G.B., R.A., D.Z. and D.B. helped with data collection; I.G. refined and analyzed the structure; J.-P.M. and C.F. designed and performed MD; N.Y., R.R., F.R.-V., E.K., D.Z. and D.B. performed meta-genomic search, discovered new viral rhodopsin sequences, made sequence alignment and phylogenetic trees; V.B., D.J.M., T.B., I.C., E.B., F.R.-V., C.F., E.K. and G.B. helped with data analysis, V.G. supervised the project; D.B., K.K. and V.G. analyzed the results and prepared the manuscript with input from all the other authors.

## Competing interests

The authors declare no competing interests.
