## [Peer Review File · Nature Communications]

Reviewers' Comments:

Reviewer #1:

Remarks to the Author:

The manuscript "Unique structure and function of viral rhodopsins" by D. Bratanov and colleagues describes the crystal structure and the functional characterization of an Organic Lake Phycodnavirus rhodopsin II, which represents the largest group of viral rhodopsins. In the crystal structure, the light-gated viral rhodopsin is a pentamer consisting of five symmetrical monomer subunits. This pentamer forms a bottle-like central channel and could be a reminiscent of well-studied pentameric ligand-controlled channels. The structural data and UV/Vis investigations are very well performed. The electrophysiology can be much more detailed. The authors should give the reader more insights to the pentameric organization and the pore function of OLPVR II via some specific point mutations. Does OLPVR II only work in a pentameric organization? How reliable is this statement? Is this channel/pore with the potential positive charged Arg-filter only anion selective? The authors should test these OLPVR II features at more detail. In addition to the new structure, it is the most important statement for the function of this protein and this manuscript. Is there any meaningful explanation as to how the rhodopsins got into the viruses? In general, I think that this study is a very interesting, novel finding and structural description for the rhodopsin community.

Additional remarks:

- The summary is too long for this journal. For my feeling, sentences like "Furthermore, viral rhodopsins could find application, in particular, in optogenetics" are not needed here.
- page 3, line 65-66. "Before the year 2000, only rhodopsins from halophilic archaea have been known", would be better to use "...only microbial rhodopsin from..."
- page 4, line 105-118: This paragraph can be shifted to the methods.
- Nearly all extended data figures are not in the correct order.
- What happens to the pentameric organization when these amino acids (Glu26, Arg36, His37, Asn40, Trp203) are mutated?
- page 4, line 145-147: I can't see this statement in Figure 1C/E. Where is the distorted C-terminus of the protomer? Where is the poorly ordered C-D loop of the neighboring protomer which forms several hydrogen bonds between its basic atoms and thus stabilizes the pentamer? The authors should label both parts in Figure 1 B and E.
- The authors should better compare this structure with their own nice data on pentamerization and structure of KR2 that have just been published (DOI: 10.1126/sciadv.aav2671). How could work the communication between the retinal binding pocket and the pore function in comparison to KR2? The authors remarks here that OLPVR II dramatically differs from those of all rhodopsins.
- Figure EDF1 Sequence alignment is very difficult to read and also partially cropped.
- What happens to the central pore when Phe24, Leu28 and Arg29 are mutated?
- Is the coordination of five water molecules in a pentagon really an important feature? These water molecules only fill the space at the narrowest positive charged bottleneck of the pore? Is this a crystallographic feature?
- The hydrophobic part of the OLPVR II pore is filled with an uncharged hydrocarbon lipid molecule but not with e.g. water molecules or ions. Is this part important for a faster ion permeation after the potential selectivity filter (Arg 26 - water pentagon)?
- The authors should add a simulated annealing omitted electron density map figure for atr.
- The authors could show a figure for the helix 4 inward shift compared to bR (line 201-212). They observed an interesting interaction of Arg72 with Asn188 but also with a Tyr (EDF15). Is this a similar interaction Tyr as in other rhodopsins? Is the Asn188 interaction unique viral rhodopsins? The authors could compare other counterpart interaction partners for Arg's from the different microbial rhodopsins.
- page 7, line 214-217. Wrong figure reference is used. EDF10?
- page 7, line 218-220. The sentence needs at least one citation.
- page 8, line 241-245. How does the similar opening of the gate around Arg72 (compared to Chr2) fit in with the function of the pentamer ion pore?

- Helices should be annotated in figure EDF16 (and in several other figures).
- page 10, line 305: "protonation of the RSB counterion". What is the RSB counterion in OLPVR11? It is only mentioned once in the text here.
- page 11 (Pumping activity of OLPVR11): It would be very interesting to investigate the electrogenic properties of OLPVR11 mutants under the aspect of influencing the possible functionally relevant pentameric organization as well as Glu26 or Glu46 in the black lipid membrane setup.
- page 12, line 391-400: The whole paragraph is very speculative and the mechanism behind the (potential) pore formation/activation is unclear to me (mechanistic model figure?).

Reviewer #2:

Remarks to the Author:

The study by Bratanov, Kovalev et al. presents the crystal structure of a viral rhodopsin named Organic Lake Phycodnavirus rhodopsin II (OLPVR11). The authors further characterize structure and function of OLPVR11 by UV-VIS spectroscopy and molecular dynamics simulations. The key hypothesis is that OLPVR11 is a pentameric light-gated ion channel, most likely, specific for chloride. There is no doubt that this is a topic of major interest with possible value for novel optogenetic applications.

Experiments and simulations are carried out following state-of-the-art procedures. However, I see two major obstacles for the drawn conclusions:

I) As far as I understand the hypothesis of the pentameric structure is solely based on the crystal structure. I am not yet convinced based on the shown data that this is also the native conformation. For example the recently discovered ChR2 X-ray structure (PDB-ID: 6eid) reveals a reverse-orientating dimer, from which it is known that it does not correspond to the native state conformation. Therefore, some biochemical data supporting the pentamer hypothesis should be shown.

II) The RMSDs in Extended Data Figure 12 A and B reveal that only Replica simulation 1 without detergent is equilibrated after about 1.4 μ s. So, for evaluation only the trajectory from 1.4 to 2 μ s should be used. From which part of the simulation are the analysis made shown in Figure 6? A Figure with RMSDs of all simulations should be shown. The shown data in Extended Data Figure 12 C and D are to me not in accordance with the drawn conclusion: "indicate convergence into a stable state for the lipid-free simulation". Without detergent (D) has more red color points reflecting higher RMSD than with (C). Anyway, a correct comparison would be taking for both cases the same simulation time spans into account (ideally equilibrated trajectories of the same length for both with and without detergent). Comparing just the first 1 μ s the stability of the simulations with and without detergent look pretty much the same! Keeping this in mind, in how far are all drawn conclusions from the simulations e.g. regarding ion conduction still reliable?

Further Comments:

1) Within several figures there dashed lines indicating hydrogen bonds that do not look like actual possible hydrogen bonds, like figure 3D (Contact between R72 and Y53). These unnecessary dashed lines make the figures confusing. In Figure 2B only the actual hydrogen bonds are shown as dashed lines, highlighting the structural important network very nicely. The dashed lines in all other figures should be adapted accordingly.

2) In the summary OLPVR11 is compared to channelrhodopsin 2 (ChR2) that both have in common to be "weak proton pumps". This is true but throughout the whole manuscript OLPVR11 is mainly compared with bacteriorhodopsin (BR). It would reflect the content of the paper in a better way to give in the summary a comparison to BR instead of ChR2.

3) Important mechanistical bacteriorhodopsin work is not cited, see for example in the review by Gerwert et al., *Biochimica et Biophysica Acta*, 2014.

4) Line 179: Do you mean Extended Data Figure 19?

Reviewer #3:

Remarks to the Author:

While the production, purification and crystallization of microbial rhodopsins is routine and robust at this stage, it is not enough to simply refer back to an earlier publication for full technical details. Sufficient detail should be provided for an interested reader to repeat the work. In this spirit then the following comments are provided and changes to the manuscript are suggested.

Page 4.

"For crystallization and functional tests, we used mostly the monomeric fraction although multimers did not show substantial differences in the experiments."

This statement is ambiguous. State exactly what was used for each experiment and why. What 'differences' are being referred to? What does 'not substantial' mean?

Page 21.

The grade and type of DDM should be reported.

Overnight should be replaced with a more exact time.

Full details regarding ultracentrifugation (time, temperature, x g) and Ni-NTA column chromatography (size, wash solution details, volumes) should be reported.

Report how the protein was concentrated before and after Superdex.

Indicate i) the exact SEC fractions pooled, ii) which were used for crystallization, and iii) which were used for each of the functional characterizations.

In what form and how was the concentrated protein stored until used for crystallization and functional work?

Report on the purity of the sample used for crystallization and functional work. Show SDS-PAGE, native gels.

Page 24.

State exactly what is meant by 'crystallization buffer'. Distinguish it from what is usually referred to as the precipitant solution.

Full details of the crystallization plate should be provided (well diameter, height).

Additional points.

Provide evidence that the pentamer is the functional form of the protein in vivo.

Provide a role for the densities identified as lipid/detergent in the central cavity. Is it possible these play no functional role and are simply a consequence of the structure determination method?

Reviewer #4:

Remarks to the Author:

Microbial rhodopsins are universal and the most abundant biological light energy transducers. Light-induced retinal isomerization causes conformational changes that within a photocycle are used for temporary channel opening in channelrhodopsins, to pump ions in pumping rhodopsins, and for signal transfer to transducers in case of sensory rhodopsins. Recently, rhodopsins in viruses were found but structure and function of these photoreceptors are unknown. The manuscript by Bratanov et al. presents the structure and functional characterization of an Organic Lake Phycodnavirus rhodopsin II (OLPVRII), a representative of the largest group of viral

rhodopsins. The authors present a high resolution structure of the viral rhodopsin which forms a pentamer as seen for other rhodopsins, like for example KR2. Interestingly and new is that the pentamer forms a bottle-like central channel. OLPVR11 forms intermolecular contacts that are unusual for microbial rhodopsins and explains forming a pore with a 5Å constriction. The authors carefully compare the OLPVR11 monomer with other microbial rhodopsins including bR, proteorhodopsin, ESR, NpSR11, KR2 and ChR2 and find for the organization of the pore in the pentamer analogies to the pentameric ion channel GLIC, where a ring of charged amino acids serves as selectivity filter. The authors show that OLPVR11 is a weak proton pump and they perform molecular dynamics simulations to get insight how the OLPVR11 pentamer could serve as a light gated ion channel. Additional spectroscopy yield information on the absorption spectra, photo cycle and pKa of Schiff base and proton acceptor.

The study is carefully done and provides new insight into viral rhodopsins, especially in a potential new function as light gated ion channel. As such it is interesting to the rhodopsin and ion channel field.

However, the study falls short in one point. For proton pumping measurements OLPVR11 has been reconstituted into E. coli polar lipids in black lipid membranes and liposomes. What is the form of the OLPVR11 in these membranes? Monomeric or pentameric? Because the purpose was to measure proton pumping activity, only TRIS and HEPES ions were present. Given that the authors suggest OLPVR11 may serve as a light gated ion channel, the authors should test different ions and proof this hypothesis.

Additional comments:

For photocycle experiments the authors state that they measured monomers and oligomers in three different environments: detergent, reconstituted in soy bean lipids and nanodiscs. They also measured photocycle of detergent solubilized monomers at pH 6, 7 and 8. In all cases they did not observe any significant photocycle differences. This is surprising. The authors should discuss this. For nanodisc assembly a ratio of DMPC:MSP1E3:OLPVR11 = 100:2:3 was used. How many rhodopsins were reconstituted per nanodisc?

The crystals were formed using monomeric OLPVR11. Is there another way to show that OLPVR11 forms pentamers in the expression host?

It would be good to add a line in the crystallography table for the number of reflections used in the Rfree set.

Reviewer #1 (Remarks to the Author):

The manuscript "Unique structure and function of viral rhodopsins" by D. Bratanov and colleagues describes the crystal structure and the functional characterization of an Organic Lake Phycodnavirus rhodopsin II, which represents the largest group of viral rhodopsins. In the crystal structure, the light-gated viral rhodopsin is a pentamer consisting of five symmetrical monomer subunits. This pentamer forms a bottle-like central channel and could be a reminiscent of well-studied pentameric ligand-controlled channels. The structural data and UV/Vis investigations are very well performed. The electrophysiology can be much more detailed.

We thank the Reviewer very much for the valuable comments which helped us to improve the manuscript. We answer the questions point by point. We also performed additional experiments favoring biological relevance of pentamer organization of OLPVR II.

The authors should give the reader more insights to the pentameric organization and the pore function of OLPVR II via some specific point mutations.

In accordance with the Reviewer suggestions we performed additional experiments. We performed crosslinking experiments with wild type protein and demonstrated that multimeric fractions consist of pentamers. In addition, the mutant E26A/R36A/W203A with the mutations of highly conserved amino acids, which are placed at the interface between the protomers and form hydrogen bonds between them, was expressed and purified. Using gel-filtration we showed that in the triple mutant the formation of the pentamer is compromised. Moreover, we reconstituted the monomers of solubilized wild type protein into liposomes, and crosslinking experiments showed that in these liposomes OLPVR II protein is in pentameric form. We changed the Extended Data Figure 6 (revised manuscript) accordingly and added the text "To prove that the observed pentameric assembly is not an artifact of crystallization, we performed crosslinking experiments with wild type protein. Using size exclusion chromatography, the protein was separated into two fractions corresponding to "monomers" and "pentamers" (Extended Data Figure 6A). Crosslinking experiments clearly confirm that multimeric fractions appearing on the size exclusion chromatography comprise of pentamers (Extended Data Figure 6B, lane 4). In addition, we expressed and purified a triple mutant E26A/R36A/W203A with three mutations of highly conserved amino acids, which are placed at the interface between the protomers and form hydrogen bonds between them. While the wild type protein demonstrated the pentamer:monomer ratio being 4:1 in the elution profile, the triple mutant showed about 1:1 ratio (Extended Data Figure 6A). Moreover, we reconstituted the monomers of solubilized wild type protein into liposomes, and crosslinking experiments showed that in these liposomes OLPVR II protein is in pentameric form (Extended Data Figure 6B, lane 4). It means that the pentamers retain their oligomeric form during purification, and monomers devoid of detergent assemble into the pentamers again, particularly during crystallization. Taking into account that the key amino acids forming the pentamer are highly conserved in the viral rhodopsins of the second group (Extended Data Figure 1), we suppose the pentameric assembly to be crucial for OLPVR II function." to the manuscript (page 5, lines 12-27, revised manuscript) to illustrate these experiments. We have also modified the corresponding section of the methods on page 24, lines 2-21.

Finally, we significantly extended our atomistic molecular dynamics simulations of wild type and various OLPVR II mutants (Figure 6 and Extended Data Figure 22). We added the following text to the manuscript: "Importantly, free-energy calculations demonstrate that charge reversal via the R29E mutation converts OLPVR II into a closed cation-selective channel, which is conductive in the F24A/L28A/R29E mutant, suggesting a similar gating mechanism as in the wild type protein (Extended Data Figure 23)." (page 13, line 2-5). Free-energy calculations reveal that OLPVR II represents an anion-selective channel, with Arg29 being the key selectivity determinant, and we find that the R29E substitution converts OLPVR II into a cation-selective channel. Based on our

simulations, our crystal structure represents a sterically open, but functionally closed state of the channel, which is regulated by a hydrophobic gating mechanism mediated by the Phe24 and Leu28 side chains.

Does OLPVR_{II} only work in a pentameric organization? How reliable is this statement?

In our work, we suppose that OLPVR_{II} functions as a pentameric anion channel with a central pore serving as a selective vestibule channel. There are three major evidences in favor of the hypotheses. First of all, it is the presence of highly conserved amino acids which are placed at the interface of the promoters and create the hydrogen bonds between them to assemble the pentamers. We have proved with crosslinking that most of the solubilized protein is in the pentameric form, and when the key amino acids are mutated the formation of the pentamers is considerably compromised. This is a strong evidence in favor of biological relevance of the pentamer organization of OLPVR_{II}. Second, the structure of the pentamer pore vestibule is similar to that of famous pentameric ligand gated channels. Even the hydrophobic tale, presumably originated from monoolein, which is host crystallization matrix, or detergent, that close the hydrophobic part of the vestibule is an analog of known drugs targeting human ligand gated pentameric channels) acts in the same way as the corresponding drugs (Mol Simul., 2014, 40(10-11): 821–829). Third, our MD simulations illustrate that the pentameric arrangement forms a central ion conduction pathway with high structural stability of the tertiary and quaternary structure on the microsecond timescale (Extended Data Figure 22).

Is this channel/pore with the potential positive charged Arg-filter only anion selective?

Yes, our MD simulations and free-energy calculations clearly suggest that OLPVR_{II} is anion selective (Figure 6 and Extended Data Figure 22). We demonstrate that the OLPVR_{II} channel is in a functionally closed state due to a hydrophobic gating mechanism mediated by F24 and L28. Channel opening in the dark-adapted state can be additionally induced by the alanine substitution of these gating residues and leads to spontaneous Cl⁻ permeation in NaCl solution. R29E substitution causes a selectivity reversal as indicated by ion free-energy profiles, and permits spontaneous Na⁺ permeation when combined with the hydrophobic gate mutation F24A/L28A.

This is consistent with the ligand gated pentameric channels (Extended Data Figure 10). Cation and anion ligand gated channels differ in type of charged amino acids covering the vestibule. For example, a putative selection filter comprising five arginine residues Arg29 means anion selectivity, but the presence of Glu means cationic nature of the channel (The EMBO Journal, 2013, 32:728–741).

The authors should test these OLPVR_{II} features at more detail. In addition to the new structure, it is the most important statement for the function of this protein and this manuscript.

Is there any meaningful explanation as to how the rhodopsins got into the viruses?

Viral rhodopsins comprise a distinct clade of microbial rhodopsin superfamily phylogenetically related to proteorhodopsin and heliorhodopsin subfamilies (Science, 2000, 289 (5486):1902-1906 and Nature, 2018, 558:595–599). Viral rhodopsins genes were initially found in genome of Nucleocytoplasmic Large DNA Viruses in Antarctica, and after that in metagenomic sequences all across the world ocean in both viruses and their algal hosts (Biology Direct, 2012, 7:34 and FEMS Microbiology Ecology, 2017, 93:fiw216). We speculate that viral rhodopsins can be subjected to coevolution between giant viruses and algae and transferred between each other via horizontal gene transfer. Many viruses and nucleofags are known to carry photosynthetic genes that are responsible for supplementing energy-generating system of the host (Nature, 2005, 438:86–89;

Nature, 2009, 461:258-262 and *Current Opinion in Virology*, 2017, 25:16–22). At the moment, exact role of viral rhodopsins in the lifecycle of giant viruses or their role during the infection remain unclear and require additional *in vivo* experiments.

In general, I think that this study is a very interesting, novel finding and structural description for the rhodopsin community.

Additional remarks:

- The summary is too long for this journal. For my feeling, sentences like "Furthermore, viral rhodopsins could find application, in particular, in optogenetics" are not needed here.

*We thank the Reviewer for the help to improve the readability of our manuscript, particularly, the Summary. We have revised the Summary, and shortened it where possible. Unfortunately, as our structure-based hypothesis is the functioning of OLPVRII as a pentameric light-gated ion channel, we suggest that it would be of high interest for further *in vivo* functional investigations, leading to the development of new OLPVRII-based optogenetic tools. Indeed, the tuning of the selectivity of such pentameric tool would be more controllable than that of other microbial rhodopsins, presumably due to the ease of the OLPVRII central pore structure (in comparison to the inner ion pathways in known light-driven pumps and channels). Thus, we could not exclude the sentence "Furthermore, viral rhodopsins could find application, in particular, in optogenetics" from the summary.*

- page 3, line 65-66. "Before the year 2000, only rhodopsins from halophilic archaea have been known", would be better to use "...only microbial rhodopsin from..."

Corrected as suggested by the Reviewer

- page 4, line 105-118: This paragraph can be shifted to the methods.

We moved the paragraph to the methods.

- Nearly all extended data figures are not in the correct order.

We numbered all Extended Data Figures according to their appearance in the text.

- What happens to the pentameric organization when these amino acids (Glu26, Arg36, His37, Asn40, Trp203) are mutated?

This is an important question. As it was mentioned above, we expressed the triple mutant E26A/R36A/W203A and observed with size exclusion chromatography remarkable decrease of the pentamer formation. We added corresponding Extended Data Figure 6 and the text "To prove that the observed pentameric assembly is not an artifact of crystallization, we performed crosslinking experiments with wild type protein. Using size exclusion chromatography, the protein was separated into two fractions corresponding to "monomers" and "pentamers" (Extended Data Figure 6A). Crosslinking experiments clearly confirm that multimeric fractions appearing on the size exclusion chromatography comprise of pentamers (Extended Data Figure 6B, lane 4). In addition, we expressed and purified a triple mutant E26A/R36A/W203A with three mutations of highly conserved amino acids, which are placed at the interface between the protomers and form

hydrogen bonds between them. While the wild type protein demonstrated the pentamer:monomer ratio being 4:1 in the elution profile, the triple mutant showed about 1:1 ratio (Extended Data Figure 6A). Moreover, we reconstituted the monomers of solubilized wild type protein into liposomes, and crosslinking experiments showed that in these liposomes OLPVR_{II} protein is in pentameric form (Extended Data Figure 6B, lane 4). It means that the pentamers retain their oligomeric form during purification, and monomers devoid of detergent assemble into the pentamers again, particularly during crystallization. Taking into account that the key amino acids forming the pentamer are highly conserved in the viral rhodopsins of the second group (Extended Data Figure 1), we suppose the pentameric assembly to be crucial for OLPVR_{II} function.” to the manuscript (page 5, lines 12-27, revised manuscript) to illustrate these experiments. We have also modified the corresponding section of the methods on page 24, lines 2-21.

- page 4, line 145-147: I can't see this statement in Figure 1C/E. Where is the distorted C-terminus of the protomer? Where is the poorly ordered C-D loop of the neighboring protomer which forms several hydrogen bonds between its basic atoms and thus stabilizes the pentamer? The authors should label both parts in Figure 1 B and E.

We have modified the figure and marked the C-D loop and C-terminus in Figure 1 B and E. In addition, we added a new Extended Data Figure 24, showing the structural alignment of the OLPVR_{II} and bR protomers. Black arrows indicate the most notable differences, including those mentioned in the text.

- The authors should better compare this structure with their own nice data on pentamerization and structure of KR2 that have just been published (DOI: 10.1126/sciadv.aav2671). How could work the communication between the retinal binding pocket and the pore function in comparison to KR2? The authors remarks here that OLPVR_{II} dramatically differs from those of all rhodopsins.

We thank the Reviewer for this comment. The Extended Data Figure 7 of the revised manuscript shows the overall architectures of the pentamers of OLPVR_{II}, KR2 and HOT75BPR. Moreover, the profiles of the central pores of different microbial rhodopsin oligomers are compared in the Extended Data Figure 8. To explain better why OLPVR_{II} pentamer is notably different from other microbial rhodopsins pentamers, we added the following text to the description of the OLPVR_{II}: “Although the relative orientation of the protomers is similar for OLPVR_{II}, KR2 and BPR HOT75 (Extended Data Figure 7), the size and profile of the central pore together with the pentamerization contacts are completely different in these proteins (Extended Data Figure 8).” on page 5, lines 28-31.

We suggest that the communication between the retinal binding pocket and the central pore is at the moment a characteristic feature of only OLPVR_{II} and consequently all viral rhodopsins of group 2. This is the result of (i) unusual architecture of the cytoplasmic region of OLPVR_{II} in comparison to known microbial rhodopsins, where the putative proton donor is Glu42 and located in helix B, which is further connected via a chain of hydrogen bonds almost to the vestibule of the central pore and (ii) unusual oligomerization contacts, which are also connected to the central pore vestibule by the dense hydrogen bonding network. Both features are described in details in the manuscript and are absent in KR2. The mechanism of sodium pumping by KR2 was demonstrated through the protein protomer, but not through the central pore, therefore we conclude that the communication between the retinal binding pocket and central pore works only for OLPVR_{II}, but not for KR2.

- Figure EDF1 Sequence alignment is very difficult to read and also partially cropped.

We changed Extended Data Figure 1 to improve the readability of the sequence alignment and eliminated the accidental crop. The figure legend was changed accordingly.

- What happens to the central pore when Phe24, Leu28 and Arg29 are mutated?

Our molecular dynamics demonstrate that the central ion pore in our OLPVR II structure is sterically open, but functionally closed, regulated by a hydrophobic gating mechanism. The main gating residues are Phe24 and Leu28. Alanine substitution of these side chains causes an additive increase in pore hydration (Figure 6f), which lowers the free-energy barrier to Cl⁻ ions and permits Cl⁻ permeation (Extended Data Figures 22). We now include additional simulations of the R29E mutation, which converts OLPVR II into a cation-selective channel, as comes from MD, regulated by the same hydrophobic gating mechanism. These data thus emphasize the key role of the Arg29 side chain in ion selectivity, but also illustrate the ability of OLPVR II to accommodate selectivity-altering mutations.

We believe that further investigations of the protein in vivo (HEK cells, neurons) will support our structure- and MD-based hypothesis of the OLPVR II function, leading to the development of the new OLPVR II-based optogenetic tools.

- Is the coordination of five water molecules in a pentagon really an important feature? These water molecules only fill the space at the narrowest positive charged bottleneck of the pore? Is this a crystallographic feature?

We thank the Reviewer for this important question. We believe, that 5 water molecules in the central pore vestibule may play important role in the ion selectivity of the OLPVR II. Our MD simulations additionally indicate that the water pentagon is stable in the structure, which may mean either that these water molecules stabilize the pentamer in its closed state (present model of OLPVR II), or that they are involved in hydration shell of the translocated ion, or both together. Indeed, MD of the prokaryotic pentameric ligand-gated ion channel GLIC shows that water pentagons are present in the transmembrane part of the channel in both closed and opened states, which means their involvement into the ion selectivity of the protein (The EMBO Journal, 2013, 32:728–741). We agree with the Reviewer that it is an important point and therefore we modified in the text the following sentence “Even more notably, the crystal structure of GLIC (PDB ID: 4HFI) contains similar pentagons of water molecules inside the transmembrane channel, presumably involved into hydration of the permeation ion, that are followed by the extremely hydrophobic region plugged with the carbon tails of detergent molecules (The EMBO Journal, 2013, 32:728–741)”.

- The hydrophobic part of the OLPVR II pore is filled with an uncharged hydrocarbon lipid molecule but not with e.g. water molecules or ions. Is this part important for a faster ion permeation after the potential selectivity filter (Arg 26 - water pentagon)?

Our MD simulations of hydrocarbon chain-free OLPVR II reveal a hydrophobic gating mechanism mediated by Phe24 and Leu28 (Figure 6 and Extended Data Figure 22). In simulations of hydrocarbon chain-bound OLPVR II, the chains are observed to frequently interchange positions, but to form a well-defined density in the pore centre in close contact with Phe24 and Leu28. Thus, the presence of these lipid tails may represent an artifact of crystallization that becomes evident in the hydrophobic gating region. Nevertheless, it is tempting to speculate that binding of similar lipid chains may modulate OLPVR II function by stabilizing the closed state of the hydrophobic gate.

- The authors should add a simulated annealing omitted electron density map figure for atr.

We built the simulated annealing omitted electron density map for the region of all-trans retinal (ATR) and covalently bound residue Lys195. We included the figure demonstrating these maps into the Extended Data Figure 4D of the revised manuscript with the examples of the electron density maps of the OLPVR II structure. We modified the figure legend accordingly.

- The authors could show a figure for the helix 4 inward shift compared to bR (line 201-212). They observed an interesting interaction of Arg72 with Asn188 but also with a Tyr (EDF15). Is this a similar interaction Tyr as in other rhodopsins? Is the Asn188 interaction unique viral rhodopsins? The authors could compare other counterpart interaction partners for Arg's from the different microbial rhodopsins.

We added a new Extended Data Figure 12 into the revised manuscript, showing the structural alignment of the OLPVRII and bR protomers. Black arrows indicate the most significant differences, including those mentioned in the text.

Unfortunately, as it was nicely noticed by another Reviewer, some of hydrogen bonds were shown in several figures by mistake (Figure 3, 4; Extended Data Figures 14-17). These include the hydrogen bond between Arg72 and Tyr53 (we also annotated this residue in Extended Data Figure 15). We also double-checked all the shown hydrogen bonds and they are correct. The Tyr53 is the analogue of Tyr57 in BR, which is also present in proteorhodopsins and sensory rhodopsins (NpSRII and ASR), however absent in sodium pumps (KR2) and ChR2.

The Asn188-Arg72 interaction is unique for viral rhodopsins of group 2. We have compared the region of Arg72 of OLPVRII, bR and ChR2 and the comparison is shown in the Extended Data Figure 15 (revised version). To stress the importance of the unusual stabilization of Arg72 in OLPVRII we added the following text to the corresponding part of the manuscript: "While Arg82 was demonstrated to be a key element in the proton translocation mechanism in bR²⁷, and analogous arginine is found in most microbial rhodopsins, playing important roles in their functioning²⁸⁻³⁰, we suggest that the strong stabilization of Arg72 in OLPVRII by three Asn side chains may affect its mobility and thus affect the function of the rhodopsin."

- page 7, line 214-217. Wrong figure reference is used. EDF10?

The figure reference was corrected to Extended Data Figure 13.

page 7, line 218-220. The sentence needs at least one citation

The references to Man et al., 2003 and Sudo and Spudich, 2006 were added.

- page 8, line 241-245. How does the similar opening of the gate around Arg72 (compared to Chr2) fit in with the function of the pentamer ion pore?

We thank the Reviewer for this fair question. Shortly, we suggest that the opening of the pore certainly is connected with the activation of the protomers, but not necessarily with the opening of a gate via Arg72. The structural basis for both protomers activation and pore opening remains unknown and require further investigation using cryo-trapping of the OLPVRII intermediates, time-resolved structural studies at XFELs or synchrotron sources, and other techniques. As the central pore and the vestibule are formed mainly by helix A, at the moment we may speculate, that pore opening is regulated by the helix A movements (shift or/and twist). We suggest that the displacement of the helix A is caused by the structural rearrangements in the OLPVRII protomers upon absorption of the photon, which propagates from the RSB via the hydrogen bond network in the cytoplasmic part of the protein from to the central pore vestibule. However, the involvement of the extracellular part, particularly, Arg72, in that process is unclear. The analysis of the RSB region and the extracellular part of the OLPVRII allow us to suggest that Arg72, similarly to the Arg82 in BR, serves as a gate for the proton translocation. The reason, why we compare OLPVRII here also with ChR2 is that the flip of the Arg72 side chain (e.g. opening of the gate) may lead to the direct connection between the RSB region and the extracellular bulk, in the same manner as it occurs in ChR2.

- Helices should be annotated in figure EDF16 (and in several other figures).

We have annotated helices in Extended Data Figures 14 and 16 and modified the figure legends correspondingly.

- page 10, line 305: “protonation of the RSB counterion”. What is the RSB counterion in OLPVRII? It is only mentioned once in the text here.

The suggestion of the Reviewer is taken into account. We changed the phrase to “RSB counterion Asp75”.

- page 11 (Pumping activity of OLPVRII): It would be very interesting to investigate the electrogenic properties of OLPVRII mutants under the aspect of influencing the possible functionally relevant pentameric organization as well as Glu26 or Glu46 in the black lipid membrane setup.

In our work we suppose that OLPVRII functions as a pentameric anion channel with a central pore serving as a selective vestibule channel. As we indicated, here are three major evidences in favor of the hypotheses. First of all, the pentameric assembly is supported by the presence of highly conserved amino acids which are placed at the interface of the promoters and create the hydrogen bonds between them. In the revised version we demonstrate, that OLPVRII forms pentamers not only in detergent and crystals, but also in the lipid membrane. Second, the structure of the pentamer pore vestibule is similar to that of famous pentameric ligand gated channels. Third, our MD simulations illustrate that the pentameric arrangement forms a central ion conduction pathway with high structural stability of the pentamer (Extended Data Figure 22). The proton pumping activity is fairly weak in OLPVRII. Together with the unusual pentameric organization and also the pore vestibule, this allows us to suggest that translocation of the protons is not the main function of OLPVRII, but is a result of inherent proton release from the RSB after photon absorption. Unfortunately, we could not prove experimentally directly that OLPVRII is an ion channel, because the current BLM setup do not allow us to observe the channeling effect and the protein is not expressed in plasma membranes of HEK cells for the patch-clamp experiments. Therefore, at the moment we do not have the opportunity to investigate how the mutations of crucial amino acids influence the activity of OLPVRII and believe, that our work on the viral rhodopsin will open the way for further studies of its electrogenic properties.

However, we simulated several most important mutations. Our molecular dynamics demonstrate that the central ion pore in our OLPVRII structure is sterically open, but functionally closed, regulated by a hydrophobic gating mechanism. The main gating residues are Phe24 and Leu28. Alanine substitution of these side chains causes an additive increase in pore hydration (Figure 6f), which lowers the free-energy barrier to Cl⁻ ions and permits Cl⁻ permeation (Extended Data Figure 22). We now include additional simulations of the R29E mutation, which converts OLPVRII into a cation-selective channel, regulated by the same hydrophobic gating mechanism.

- page 12, line 391-400: The whole paragraph is very speculative and the mechanism behind the (potential) pore formation/activation is unclear to me (mechanistic model figure?).

We added to the supplements mechanistic model figure (Extended Data Figure 24) and the text “In Extended Data Figure 24 we present a simple mechanistic model of OLPVRII anion channeling function.” on page 13, lines 7-9.

Reviewer #2 (Remarks to the Author):

The study by Bratanov, Kovalev et al. presents the crystal structure of a viral rhodopsin named Organic Lake Phycodnavirus rhodopsin II (OLPVR_{II}). The authors further characterize structure and function of OLPVR_{II} by UV-VIS spectroscopy and molecular dynamics simulations. The key hypothesis is that OLPVR_{II} is a pentameric light-gated ion channel, most likely, specific for chloride. There is no doubt that this is a topic of major interest with possible value for novel optogenetic applications.

Experiments and simulations are carried out following state-of-the-art procedures. However, I see two major obstacles for the drawn conclusions.

D) As far as I understand the hypothesis of the pentameric structure is solely based on the crystal structure. I am not yet convinced based on the shown data that this is also the native conformation. For example the recently discovered ChR2 X-ray structure (PDB-ID: 6eid) reveals a reverse-orientating dimer, from which it is known that it does not correspond to the native state conformation. Therefore, some biochemical data supporting the pentamer hypothesis should be shown.

We acknowledge the Reviewer comments helping us to improve the manuscript.

It is a useful remark. To prove the physiological relevance of OLPVR_{II} pentameric organization we performed crosslinking experiments with wild type protein and demonstrated that multimeric fractions consist of pentamers. In addition, the mutant E26A/R36A/W203A with the mutations of highly conserved amino acids, which are placed at the interface between the protomers and form hydrogen bonds between them, was expressed and purified. Using gel-filtration we showed that in the triple mutant the formation of the pentamer is compromised. Moreover, we reconstituted the monomers of solubilized wild type protein into liposomes, and crosslinking experiments showed that in these liposomes OLPVR_{II} protein is in pentameric form. We changed the Extended Data Figure 6 (revised manuscript) accordingly and added the text “To prove that the observed pentameric assembly is not an artifact of crystallization, we performed cross-linking experiments with wild type protein. Using size exclusion chromatography, the protein was separated into two fractions corresponding to “monomers” and “pentamers” (Extended Data Figure 6A). Crosslinking experiments clearly confirm that multimeric fractions appearing on the size exclusion chromatography comprise of pentamers (Extended Data Figure 6B, lane 4). In addition, we expressed and purified a triple mutant E26A/R36A/W203A with three mutations of highly conserved amino acids, which are placed at the interface between the protomers and form hydrogen bonds between them. While the wild type protein demonstrated the pentamer:monomer ratio being 4:1 in the elution profile, the triple mutant showed about 1:1 ratio (Extended Data Figure 6A). Moreover, we reconstituted the monomers of solubilized wild type protein into liposomes, and crosslinking experiments showed that in these liposomes OLPVR_{II} protein is in pentameric form (Extended Data Figure 6B, lane 4). It means that the pentamers retain their oligomeric form during purification, and monomers devoid of detergent assemble into the pentamers again, particularly during crystallization. Taking into account that the key amino acids forming the pentamer are highly conserved in the viral rhodopsins of the second group (Extended Data Figure 1), we suppose the pentameric assembly to be crucial for OLPVR_{II} function.” to the manuscript (page 5, lines 12-27, revised manuscript) to illustrate these experiments. We have also modified the corresponding section of the methods on page 24, lines 2-21.

In addition, we would like to mention, that in case of the ChR2 X-ray structure (PDB ID: 6EID), two slightly different ChR2 reverse-oriented molecules are present in the asymmetric unit, as it was mentioned by the Reviewer. However, each of these molecules is a part of the dimer in the crystal lattice, corresponding to the native state conformation. The dimers were identified and shown in crystals in the corresponding publication. Thus it was shown, that ChR2 is in its native state in

crystals. In case of OLPVRII the pentamer is present in the asymmetric unit, and no more oligomers were found in the crystal lattice using symmetry operators (Extended Data Figure 4).

II) The RMSDs in Extended Data Figure 12 A and B reveal that only Replica simulation 1 without detergent is equilibrated after about 1.4 μ s. So, for evaluation only the trajectory from 1.4 to 2 μ s should be used. From which part of the simulation are the analysis made shown in Figure 6? A Figure with RMSDs of all simulations should be shown. The shown data in Extended Data Figure 12 C and D are to me not in accordance with the drawn conclusion: “indicate convergence into a stable state for the lipid-free simulation”. Without detergent (D) has more red color points reflecting higher RMSD than with (C). Anyway, a correct comparison would be taking for both cases the same simulation time spans into account (ideally equilibrated trajectories of the same length for both with and without detergent). Comparing just the first 1 μ s the stability of the simulations with and without detergent look pretty much the same! Keeping this in mind, in how far are all drawn conclusions from the simulations e.g. regarding ion conduction still reliable?

We agree that the backbone RMSDs calculated for the pentamer shown in the previous version of the manuscript could indeed indicate insufficient convergence. Nevertheless, the RMSDs against the crystal structure were below 2 Å for up to 1 μ s suggesting relatively high structural stability of the simulation system.

To address any potential concern regarding insufficient sampling and structural stability, we extended all simulations up to 2 μ s and started additional independent simulation replicas for each condition, thereby now reaching an aggregated simulation time of ~30 μ s. In Extended Data Figure 23 of the revised manuscript, we now show an overview of RMSD time courses against our crystal structure for all simulations calculated (a) for both transmembrane backbones and (b) for the narrowest part of the ion pore. These RMSDs are now sufficiently converged, both in the time domain and across replicas, indicating high structural stability of the pentameric channel and in particular of the central ion conduction pathway. In the Methods section, we now clearly state that all analyses of these simulations for Figure 6 are consistently based on the time windows from 1400–1800 ns of each simulation.

Our conclusions regarding the structural stability with/without bound hydrocarbon tails based on all-to-all RMSD maps was apparently misleading. We therefore removed this part. Based on our new simulations, we now conclude that the pentamer is similarly stable in absence and presence of hydrocarbon tail bound.

Further Comments:

1) Within several figures there dashed lines indicating hydrogen bonds that do not look like actual possible hydrogen bonds, like figure 3D (Contact between R72 and Y53). These unnecessary dashed lines make the figures confusing. In Figure 2B only the actual hydrogen bonds are shown as dashed lines, highlighting the structural important network very nicely. The dashed lines in all other figures should be adapted accordingly.

We thank the Reviewer for this very important comment. We agree that some hydrogen bonds were shown with black dashed lines by mistake. We deleted those black dashed lines in several figures (Figure 3, 4; Extended Data Figures 14-17).

2) In the summary OLPVRII is compared to channelrhodopsin 2 (ChR2) that both have in common to be "weak proton pumps". This is true but throughout the whole manuscript OLPVRII is mainly compared with bacteriorhodopsin (BR). It would reflect the content of the paper in a better way to give in the summary a comparison to BR instead of ChR2.

Following the suggestion of another Reviewer, we have shortened the Summary and improved its readability. As it is stated in the Summary: “The architecture of the OLPVRII, with the central

pore, is unique among the rhodopsins of known structure”, OLPVRII indeed differs from both bacteriorhodopsin (BR) and channelrhodopsin-2 (ChR2). However, the viral rhodopsin acts as a weak proton pump, unlike BR, but similar to ChR2. The proton pumping activity is fairly weak in OLPVRII. This, together with the unusual pentameric organization and also the pore vestibule, allow us to suggest, that translocation of the protons is not the main function of OLPVRII, but is a result of inherent proton release from the RSB after photon absorption. In the text we compare OLPVRII with BR, as BR is a classical representative of microbial rhodopsins, and demonstration of the structural differences between these two proteins will be transparent for the wide audience. However, we also compared the extracellular parts of OLPVRII and ChR2 in the text, due to the similarity of Arg72 gate with ChR2. Thus, we modified the sentence in the Summary to be: “We show the protein is a weak proton pump, unlike bacteriorhodopsin, but similar to channelrhodopsin-2.”.

3) Important mechanistical bacteriorhodopsin work is not cited, see for example in the review by Gerwert et al., *Biochimica et Biophysica Acta*, 2014.

We agree with the Reviewer and corrected this in the revised manuscript. We added an additional reference on page 8, line 16.

4) Line 179: Do you mean Extended Data Figure 19?

Yes. We corrected the corresponding figure reference (Extended Data Figure 10 in the revised manuscript).

Reviewer #3 (Remarks to the Author):

While the production, purification and crystallization of microbial rhodopsins is routine and robust at this stage, it is not enough to simply refer back to an earlier publication for full technical details. Sufficient detail should be provided for an interested reader to repeat the work. In this spirit then the following comments are provided and changes to the manuscript are suggested.

Page 4.

“For crystallization and functional tests, we used mostly the monomeric fraction although multimers did not show substantial differences in the experiments.” This statement is ambiguous. State exactly what was used for each experiment and why. What ‘differences’ are being referred to? What does ‘not substantial’ mean?

We acknowledge the Reviewer comments helping us to improve the manuscript.

Concerning the differences of multimers and monomers that we observed in functional tests. On the page 23, line 6 of revised manuscript, it is indicated that “For crystallization and functional tests, we used mostly the monomeric fraction although multimers did not show substantial differences in the experiments.” We conducted additional cross-linking experiments and showed that multimeric fractions consist of pentamers. Pentamers were used only for time-resolved UV-Vis spectroscopy that is mentioned on page 25, line 29 and crosslinking experiments that is mentioned on page 5, line 15; and in the corresponding section of the methods on page 24, lines 2-21. To remove the ambiguity we shortened the corresponding sentence to “For crystallization and functional tests, we used the monomeric fraction unless otherwise indicated.” However, we added the sentence “The photocycle of the monomers in detergent was shorter, but the order and appearance of the intermediates were the same for all samples.” on page 25, lines 31-33 to indicate what differences were observed.

The close features of the photocycles measured under different conditions are quite unusual, but not unique. The sensitivity of rhodopsin photocycle towards lipid/detergent environment may be different. Whereas the replacement of the native lipid matrix by detergents significantly changes the kinetics of bR, in opposite, the photocycles of pHR and pSRII were found almost identical upon solubilization in DM (Biophysical Journal, 2001, 81:1600-1612; Biophysical Journal, 1998, 75:999-1009). So far, this has not been reasonably well explained in the literature, therefore we leave our observations as a fact and only add the following text to comment this result: “Close features of the photocycles measured under different conditions are quite unusual, but not unique. Whereas the replacement of the native lipid matrix by detergents significantly changes the kinetics of bR, in opposite, the photocycles of pHR and pSRII were found almost identical upon solubilization in DM^{58,59}.” on page 25, line 34.

Page 21.

The grade and type of DDM should be reported. Overnight should be replaced with a more exact time.

Full details regarding ultracentrifugation (time, temperature, x g) and Ni-NTA column chromatography (size, wash solution details, volumes) should be reported.

Report how the protein was concentrated before and after Superdex.

Indicate i) the exact SEC fractions pooled, ii) which were used for crystallization, and iii) which were used for each of the functional characterizations.

In what form and how was the concentrated protein stored until used for crystallization and functional work?

Report on the purity of the sample used for crystallization and functional work. Show SDS-PAGE, native gels.

As suggested by the Reviewer the details of detergent purity, cultivation time, ultracentrifugation, metal affinity chromatography, concentration, storing, and purity of the protein were added in the corresponding part of the Materials and Methods chapter (pages 22 and 23).

The exact SEC fractions pooled are shown on Extended Data Figure 6A on page 42 of the revised manuscript.

As we wrote above, on the page 23, line 6, it was indicated that “For crystallization and functional tests, we used mostly the monomeric fraction although multimers did not show substantial differences in the experiments.” We conducted additional cross-linking experiments and showed that multimeric fractions consist of pentamers. Pentamers were used only for time-resolved UV-Vis spectroscopy that is mentioned on page 25, line 29 and crosslinking experiments that is mentioned on page 5, line 15; and in the corresponding section of the methods on page 24, lines 2-21. To remove the ambiguity we shortened the corresponding sentence to “For crystallization and functional tests, we used the monomeric fraction unless otherwise indicated.” However, we added the sentence “The photocycle of the monomers in detergent was shorter, but the order and appearance of the intermediates were the same for all samples.” on page 25, lines 31-33.

We added the SDS-PAGE of the purified samples to the Extended Data Figure 6B and the text “Typically it was possible to obtain the protein peak ratio A_{280}/A_{520} of 1.2 or lower that usually indicates a very high purity of the samples. The purity of the samples is also illustrated with SDS-PAGE (Extended Data Figure 6B, lanes 1 and 2).” on page 23, lines 9-13 to illustrate the purity of the samples for the readers. The legend of the Extended Data Figure 6 was modified accordingly.

Page 24.

State exactly what is meant by ‘crystallization buffer’. Distinguish it from what is usually referred to as the precipitant solution.

Full details of the crystallization plate should be provided (well diameter, height).

Under “protein in the crystallization buffer” we meant the purified solubilized in DDM protein after SEC in the respective buffer, typically 50 mM $\text{NaH}_2\text{PO}_4/\text{Na}_2\text{HPO}_4$ pH 8.0, 100 mM NaCl. Indeed, the words “in the crystallization buffer” are confusing, therefore we removed them from the revised manuscript. We also corrected the name of the crystallization plate to “96-well LCP glass sandwich set (Paul Marienfeld GmbH, Germany)” indicating the trade name of the corresponding product and the producer. The dimensions of the plate are well documented on the website of the producer (<https://www.marienfeld-superior.com/lcp-lipidic-cubic-phase-sandwich-set-2768.html>), therefore we did not include them, however, in the revised manuscript we provided corresponding reference to the website on page 26, line 26.

Additional points.

Provide evidence that the pentamer is the functional form of the protein *in vivo*.

To prove the physiological relevance of OLPVR II pentameric organization we performed crosslinking experiments with wild type protein and demonstrated that multimeric fractions consist of pentamers. In addition, the mutant E26A/R36A/W203A with the mutations of highly conserved amino acids, which are placed at the interface between the protomers and form hydrogen bonds between them, was expressed and purified. Using gel-filtration we showed that in the triple mutant the formation of the pentamer is compromised. Moreover, we reconstituted the monomers of solubilized wild type protein into liposomes, and crosslinking experiments showed that in these liposomes OLPVR II protein is in pentameric form. We changed the Extended Data Figure 6

(revised manuscript) accordingly and added the text “To prove that the observed pentameric assembly is not an artifact of crystallization, we performed cross-linking experiments with wild type protein. Using size exclusion chromatography, the protein was separated into two fractions corresponding to “monomers” and “pentamers” (Extended Data Figure 6A). Crosslinking experiments clearly confirm that multimeric fractions appearing on the size exclusion chromatography comprise of pentamers (Extended Data Figure 6B, lane 4). In addition, we expressed and purified a triple mutant E26A/R36A/W203A with three mutations of highly conserved amino acids, which are placed at the interface between the protomers and form hydrogen bonds between them. While the wild type protein demonstrated the pentamer:monomer ratio being 4:1 in the elution profile, the triple mutant showed about 1:1 ratio (Extended Data Figure 6A). Moreover, we reconstituted the monomers of solubilized wild type protein into liposomes, and crosslinking experiments showed that in these liposomes OLPVR II protein is in pentameric form (Extended Data Figure 6B, lane 4). It means that the pentamers retain their oligomeric form during purification, and monomers devoid of detergent assemble into the pentamers again, particularly during crystallization. Taking into account that the key amino acids forming the pentamer are highly conserved in the viral rhodopsins of the second group (Extended Data Figure 1), we suppose the pentameric assembly to be crucial for OLPVR II function.” to the manuscript (page 5, lines 12-27, revised manuscript) to illustrate these experiments. We have also modified the corresponding section of the methods on page 24, lines 2-21.

Provide a role for the densities identified as lipid/detergent in the central cavity. Is it possible these play no functional role and are simply a consequence of the structure determination method?

The strong difference electron density was identified in the hydrophobic part of the central pore vestibule which was fitted very well by the hydrocarbon chain, corresponding to either lipid or a detergent molecule. Our MD simulations of hydrocarbon chain-free OLPVR II reveal a hydrophobic gating mechanism mediated by Phe24 and Leu28 (Figure 6 and Extended Data Figure 22 of the revised manuscript). In simulations of hydrocarbon chain-bound OLPVR II, the chains are observed to frequently interchange positions, but to form a well-defined density in the pore center in close contact with Phe24 and Leu28. Thus, the presence of these lipid tails may represent an artifact of crystallization that becomes evident in the hydrophobic gating region. However, we cannot exclude that the hydrocarbon chain may originate from the purification step of the OLPVR II, when the protein is solubilized in DDM. We believe that this molecule is absent in the OLPVR II pentamers in native membranes and therefore is an artifact of the purification/crystallization steps. Similar molecules appeared in the same manner in pentameric ligand-gated ion channels (pLGICs). Indeed, the hydrophobic region of the transmembrane domain of GLIC is filled with detergent molecules, blocking the channel (The EMBO Journal, 2013, 32:728-741). Due to high similarities of the structures of membrane parts (vestibules) of pLGICs and OLPVR II we suggest similar nature of the blocking of the vestibule in the closed channel. Nevertheless, it is tempting to speculate that binding of similar lipid chains may modulate OLPVR II function by stabilizing the closed state of the hydrophobic gate.

Reviewer #4 (Remarks to the Author):

Microbial rhodopsins are universal and the most abundant biological light energy transducers. Light-induced retinal isomerization causes conformational changes that within a photocycle are used for temporary channel opening in channelrhodopsins, to pump ions in pumping rhodopsins, and for signal transfer to transducers in case of sensory rhodopsins. Recently, rhodopsins in viruses were found but structure and function of these photoreceptors are unknown. The manuscript by Bratanov et al. presents the structure and functional characterization of an Organic Lake Phycodnavirus rhodopsin II (OLPVRII), a representative of the largest group of viral rhodopsins. The authors present a high resolution structure of the viral rhodopsin which forms a pentamer as seen for other rhodopsins, like for example KR2. Interestingly and new is that the pentamer forms a bottle-like central channel. OLPVRII forms intermolecular contacts that are unusual for microbial rhodopsins and explains forming a pore with a 5Å constriction. The authors carefully compare the OLPVRII monomer with other microbial rhodopsins including bR, proteorhodopsin, ESR, NpSRII, KR2 and ChR2 and find for the organization of the pore in the pentamer analogies to the pentameric ion channel GLIC, where a ring of charged amino acids serves as selectivity filter. The authors show that OLPVRII is a weak proton pump and they perform molecular dynamics simulations to get insight how the OLPVRII pentamer could serve as a light gated ion channel. Additional spectroscopy yield information on the absorption spectra, photo cycle and pKa of Schiff base and proton acceptor.

The study is carefully done and provides new insight into viral rhodopsins, especially in a potential new function as light gated ion channel. As such it is interesting to the rhodopsin and ion channel field.

However, the study falls short in one point. For proton pumping measurements OLPVRII has been reconstituted into E. coli polar lipids in black lipid membranes and liposomes. What is the form of the OLPVRII in these membranes? Monomeric or pentameric?

We thank the Reviewer for valuable comments.

In accordance with the Reviewer suggestions (and other Reviewers as well) we performed additional experiments. We performed crosslinking experiments with wild type protein and demonstrated that multimeric fractions consist of pentamers. Moreover, we reconstituted the monomers of solubilized wild type protein into liposomes, and crosslinking experiments showed that in these liposomes OLPVRII protein is in pentameric form. We changed the Extended Data Figure 6 (revised manuscript) accordingly and added the text “To prove that the observed pentameric assembly is not an artifact of crystallization, we performed cross-linking experiments with wild type protein. Using size exclusion chromatography, the protein was separated into two fractions corresponding to “monomers” and “pentamers” (Extended Data Figure 6A). Crosslinking experiments clearly confirm that multimeric fractions appearing on the size exclusion chromatography comprise of pentamers (Extended Data Figure 6B, lane 4). In addition, we expressed and purified a triple mutant E26A/R36A/W203A with three mutations of highly conserved amino acids, which are placed at the interface between the protomers and form hydrogen bonds between them. While the wild type protein demonstrated the pentamer:monomer ratio being 4:1 in the elution profile, the triple mutant showed about 1:1 ratio (Extended Data Figure 6A). Moreover, we reconstituted the monomers of solubilized wild type protein into liposomes, and crosslinking experiments showed that in these liposomes OLPVRII protein is in pentameric form (Extended Data Figure 6B, lane 4). It means that the pentamers retain their oligomeric form during purification, and monomers devoid of detergent assemble into the pentamers again, particularly during crystallization. Taking into account that the key amino acids forming the pentamer are highly conserved in the viral rhodopsins of the second group (Extended Data Figure 1), we suppose the

pentameric assembly to be crucial for OLPVRII function.” to the manuscript (page 5, lines 12-27, revised manuscript) to illustrate these experiments. We have also modified the corresponding section of the methods on page 24, lines 2-21.

Because the purpose was to measure proton pumping activity, only TRIS and HEPES ions were present. Given that the authors suggest OLPVRII may serve as a light gated ion channel, the authors should test different ions and proof this hypothesis.

We conducted also the experiments with NaCl, KCl, and did not observe sodium, potassium, or chloride pumping. So we could conclude that the ions are not pumped by OLPVRII. The current BLM setup do not allow us to observe the channeling effect. Unfortunately, the protein is not expressed in plasma membranes of HEK cells to prove experimentally directly that OLPVRII is an ion channel. However, molecular dynamics clearly suggests that it is a chloride channel. However, there are strong evidences in favor of the hypothesis. First of all, it is the presence of highly conserved amino acids, which are placed at the interface of the promoters and create the hydrogen bonds between them to assemble the pentamers. We have proved that most of the solubilized protein is in the pentameric form, and when the key amino acids are mutated the formation of the pentamers is considerably compromised. This is a strong evidence in favor of biological relevance of the pentamer organization of OLPVRII. Second, the structure of the pentamer pore vestibule is similar to that of famous pentameric ligand gated channels. Even the hydrophobic tale, presumably originated from monoolein, which is host crystallization matrix, or detergent, that close the hydrophobic part of the vestibule is an analog of known drugs targeting human ligand gated pentameric channels) acts in the same way as the corresponding drugs (Mol Simul., 2014, 40(10-11): 821–829). Third, our MD simulations illustrate that the pentameric arrangement forms a central ion conduction pathway with high structural stability of the tertiary and quaternary structure on the microsecond timescale (Extended Data Figure 22).

Additional comments:

For photocycle experiments the authors state that they measured monomers and oligomers in three different environments: detergent, reconstituted in soy bean lipids and nanodiscs. The also measured photocycle of detergent solubilized monomers at pH 6, 7 and 8. In all cases they did not observe any significant photocycle differences. This is surprising. The authors should discuss this.

We agree with the Reviewer that close features of the photocycles measured under different conditions are quite unusual, but this phenomenon is not unique. Similarly, the bacteriorhodopsin does not show any significant changes of the photocycle within much wider range of pHs (from 5 to 9, Xie et al, Biophysical Journal, 1987, 51:627-635), as well as another retinal proteins pSRII (Biophysical Journal,1998, 75:999-1009) and proteorhodopsin (J. Mol. Biol. (2002) 321, 821–838). The sensitivity of rhodopsin photocycle towards lipid/detergent environment may be different. Whereas the replacement of the native lipid matrix by detergents significantly changes the kinetics of bR, in opposite, the photocycles of pHR and pSRII were found almost identical upon solubilization in DM (Biophysical Journal,2001, 81:1600-1612; Biophysical Journal,1998, 75:999-1009). So far, this has not been reasonably well explained in the literature, therefore we leave our observations as a fact and only add the following text to comment this result: “Close features of the photocycles measured under different conditions are quite unusual, but not unique. Whereas the replacement of the native lipid matrix by detergents significantly changes the kinetics of bR, in opposite, the photocycles of pHR and pSRII were found almost identical upon solubilization in DM^{58,59}.” on page 25, line 34.

For nanodisc assembly a ratio of DMPC:MSP1E3:OLPVRII = 100:2:3 was used. How many rhodopsins were reconstituted per nanodisc?

The nanodiscs are relatively small. According to the protocol, we should obtain the nanodiscs with the size of 12-13 nm, which is close to the size of the pentamer – around 8-9 nm. Therefore, there is only one pentamer/monomer per nanodisc. The empty nanodiscs and aggregates were removed by SEC. We added the corresponding comment to the text of the revised manuscript (page 23, line 35).

The crystals were formed using monomeric OLPVR_{II}. Is there another way to show that OLPVR_{II} forms pentamers in the expression host?

To prove the physiological relevance of OLPVR_{II} pentameric organization we performed crosslinking experiments with wild type protein and demonstrated that multimeric fractions consist of pentamers. In addition, the mutant E26A/R36A/W203A with the mutations of highly conserved amino acids, which are placed at the interface between the protomers and form hydrogen bonds between them, was expressed and purified. Using gel-filtration we showed that in the triple mutant the formation of the pentamer is compromised. Moreover, we reconstituted the monomers of solubilized wild type protein into liposomes, and crosslinking experiments showed that in these liposomes OLPVR_{II} protein is in pentameric form. We changed the Extended Data Figure 6 (revised manuscript) accordingly and added the text “To prove that the observed pentameric assembly is not an artifact of crystallization, we performed cross-linking experiments with wild type protein. Using size exclusion chromatography, the protein was separated into two fractions corresponding to “monomers” and “pentamers” (Extended Data Figure 6A). Crosslinking experiments clearly confirm that multimeric fractions appearing on the size exclusion chromatography comprise of pentamers (Extended Data Figure 6B, lane 4). In addition, we expressed and purified a triple mutant E26A/R36A/W203A with three mutations of highly conserved amino acids, which are placed at the interface between the protomers and form hydrogen bonds between them. While the wild type protein demonstrated the pentamer:monomer ratio being 4:1 in the elution profile, the triple mutant showed about 1:1 ratio (Extended Data Figure 6A). Moreover, we reconstituted the monomers of solubilized wild type protein into liposomes, and crosslinking experiments showed that in these liposomes OLPVR_{II} protein is in pentameric form (Extended Data Figure 6B, lane 4). It means that the pentamers retain their oligomeric form during purification, and monomers devoid of detergent assemble into the pentamers again, particularly during crystallization. Taking into account that the key amino acids forming the pentamer are highly conserved in the viral rhodopsins of the second group (Extended Data Figure 1), we suppose the pentameric assembly to be crucial for OLPVR_{II} function.” to the manuscript (page 5, lines 12-27, revised manuscript) to illustrate these experiments. We have also modified the corresponding section of the methods on page 24, lines 2-21.

It would be good to add a line in the crystallography table for the number of reflections used in the R_{free} set.

We added the number of reflections used for R_{free} set into the crystallography table.

Reviewers' Comments:

Reviewer #1:

Remarks to the Author:

Overall, the authors have satisfactorily addressed my comments. The manuscript "Unique structure and function of viral rhodopsins" by Bratanov and coworkers describe very nicely a novel structure of a viral rhodopsin. The functional characterization is not complete because of missing patch-clamp data. But for a first structural description this is enough for my feelings. However, I strongly suggest that in the final remarks it be clearly noted again that further experimental evidence is required for the channel function (ion conductivity/selectivity). Only further investigations of the electrogenic properties of the protein in vivo - patch clamp experiments in HEK cells or neurons - will support the hypotheses of the functioning of OLPVR11 in this manuscript. Thus, it could not be directly proven experimentally that OLPVR11 is a real ion channel and how it works. Therefore, the statement to successfully develop new OLPVR11-based optogenetic tools is still very vague.

Reviewer #2:

Remarks to the Author:

The authors provided satisfactory explanations in part to my raised concern. However a central point is not addressed. Most of the ideas presented and described here are based on earlier pioneering work on bacteriorhodopsin as for example described in ref 28. To reduce this contribution to the understanding of bacteriorhodopsins mechanism to the role of arg 82 is not correct and deserves still major correction.

I have two minor comments:

Figure 5 a: The legend in the figure seems to be wrong. I believe the red line is 1799 and monesin and the blue line is no protonophore.

Extended Data Figure 6a: Why is the pentamer peak of the mutant significantly shifted compared to wild type. Also the monomer peak is shifted a bit. This should be explained in the text.

Reviewer #3:

Remarks to the Author:

The revised manuscript is technically much improved. However, the English composition has suffered. In the interests of clarity, the manuscript will require careful editing.

Comments

52. My reading of this would lead me to believe there is direct functional evidence in the manuscript for light activated non-protonic ion channeling. The only evidence for it is from MDS. This should be made clear in the abstract.

142. Presumably there is a mixture of oligomeric states present across all rhodopsin containing fractions. In other words it is not just monomers and pentamers that are present. This should be clarified.

144. Relatedly, my understanding is that the data supports the presence of pentamers but not exclusively pentamers as possibly suggested by the text.

148. Indicate what stabilizes pentamer formation in the absence of pentamerization residues in the triple mutant.

150. Meaning pentamers are likely present. Explain why other oligomeric forms are observed? Are these physiologically relevant?

152. Where is the evidence the protein is devoid of detergent?

363. Is it not possible to tell what the yield of protein was? Was it sufficiently low to account for

the failure to detect activity or is something else amiss?
364. "could be achieved" Were they achieved?
372. Confusing. Liposomes are referred to but the opening sentence refers to BLM experiments?
387. It would be useful to define an activity scale and what 'weak' corresponds to on that scale.
399. A pore that is ~4 Å across is big enough to allow water and other substances to pass. This mean the pentamer creates an open pore in the membrane? Blockage of polar substances is proposed to occur by way of the hydrophobic gate. Does this mean the pentamer provides access to small hydrophobic substances? What might be the consequences if so?
422. How active is the channel expected to be and how does this activity compare to other channels in current optogenetics use? In other words, how realistic is the idea of using this system for optogenetics applications?
710. Spell out what DDM is. Is it alpha or beta DDM or a mix?
715. Rewrite for clarity.
726. Did the ratio change across the chromatogram from monomers to pentamers?
736. State lipid concentration.
739. Provide more details regarding detergent removal with beads.
741. State the type and size of liposomes that result and were used. Multilamellar, unilamellar, a mix?
750. Report source of lipoprotein and conditions for nanodisc formation.
755. PS contains an amino group.
756. mole or weight ratio?
784. Report the concentration of cells and liposomes used in light-induced pH change studies
819. State how much shorter.
821. Rewrite for clarity.
849. Link does not work. Provide information requested.
901. Explain how cavity was determined.
905. No certainty the fragment is lipid. It could be detergent. Suggest using hydrocarbon.
Fig 3. Explain pink volumes in this and other figures.
907. Brown arrows represent speculation and should be identified as such.
Edf 6. Explain the large shift in elution volume between the wt and the triple mutant.
945. State how pKa is determined.
952. Are these two pumping experiments directly comparable? Same liposome density, same protein density, etc? This should be clarified in the legend.
Edf 22/23 legends mixed up.
Edf 24. Does the mechanism of channel opening only require one retinal to isomerize as suggested in the scheme? What is the likelihood of multiple isomerizations?
Why only chloride channeling? Are there other possibilities? Water and protons can cross. Proton movement would collapse the membrane gradient.
How confident are you about the directionality of flow?
Is there evidence that isomerization can trigger helix A rotation? Was this/can it be modelled by MDS? Would this rotation open the hole?
What function would be served by a light driven transport of the type proposed?
Label sides of membrane.

Reviewer #4:

Remarks to the Author:

The authors addressed my critique satisfactorily. Especially they addressed the monomer/pentamer state of the rhodopsin, a concern that was also raised by other reviewers.

Reviewer #1 (Remarks to the Author):

Overall, the authors have satisfactorily addressed my comments. The manuscript “Unique structure and function of viral rhodopsins” by Bratanov and coworkers describe very nicely a novel structure of a viral rhodopsin. The functional characterization is not complete because of missing patch-clamp data. But for a first structural description this is enough for my feelings.

However, I strongly suggest that in the final remarks it be clearly noted again that further experimental evidence is required for the channel function (ion conductivity/selectivity). Only further investigations of the electrogenic properties of the protein in vivo - patch clamp experiments in HEK cells or neurons - will support the hypotheses of the functioning of OLPVRII in this manuscript. Thus, it could not be directly proven experimentally that OLPVRII is a real ion channel and how it works. Therefore, the statement to successfully develop new OLPVRII-based optogenetic tools is still very vague.

We thank the Reviewer for the comments on our manuscript. We agree that, indeed, additional experiments (particularly, patch clamp studies) are required to directly prove that OLPVRII and viral rhodopsins of group 2 serve as light-gated pentameric ion channels. To make it clear in the abstract, we modified the last sentence as following: “Additional experiments are necessary to explore potential applications of viral rhodopsins, in particular, in optogenetics.”

Reviewer #2 (Remarks to the Author):

The authors provided satisfactory explanations in part to my raised concern. However a central point is not addressed. Most of the ideas presented and described here are based on earlier pioneering work on bacteriorhodopsin as for example described in ref 28. To reduce this contribution to the understanding of bacteriorhodopsins mechanism to the role of arg 82 is not correct and deserves still major correction.

We thank the Reviewer for his comments on our revised manuscript. We agree that the reduction of the contribution to the understanding of the proton transport mechanism by bacteriorhodopsin (bR) only to the role of Arg82 is incorrect.

We revised the manuscript, particularly the parts describing OLPVRII protomer organization and its comparison with bR and referred to those pioneering works on bR and its mechanisms in the paragraphs regarding RSB deprotonation (page 8, paragraph 2) and reprotonation (page 8, paragraph 4), the role of Leu93 in the formation of water-mediated chain of hydrogen bonds between Asp96 and RSB during bR photocycle (page 9, paragraph 2).

We also stressed additionally the role of water molecules, stabilizing the Arg82 in bR on page 8, paragraph 3.

We believe, that now our manuscript is more complete and readable.

I have two minor comments:

Figure 5 a: The legend in the figure seems to be wrong. I believe the red line is 1799 and monesin and the blue line is no protonophore.

We noticed that in the last version of the picture we changed the colors of the curves, but forgot to modify the text. We corrected the text. In contrast to direct pH measurements with liposomes, addition of CCCP (1799) and monensin lead to the appearance of stationary currents in BLM experiments.

Extended Data Figure 6a: Why is the pentamer peak of the mutant significantly shifted compared to wild type. Also the monomer peak is shifted a bit. This should be explained in the text.

The relative shift of the peaks in SEC can be caused by multiple reasons or their combination, we do not know the exact answer. Possible reasons could include the mutual influence of the positions of the peaks and their heights on each other, the influence of the mutation itself, the difference of the detergent binding to the wild type and mutant proteins, differences in mobility of the samples, and different setups where the experiments were carried out. For not to be too speculative, we will refrain from the comments in the main text as, in our opinion, here the SEC profiles shows very clear the qualitative picture of the monomer:pentamer distribution, but they are not suitable for quantitative conclusions. However, we included a comment into the figure legend: “We suppose that the relative shifts of the monomers and pentamers are probably caused by a combination of several reasons, including the mutual influence of the positions of the peaks and their heights on each other, the influence of the mutation itself, the difference of the detergent binding to the wild type and mutant proteins, differences in mobility of the samples, and different setups where the experiments were carried out. Nevertheless the elution profiles clearly illustrate the distribution of the protein between monomeric and pentameric fractions.”.

Reviewer #3 (Remarks to the Author):

The revised manuscript is technically much improved. However, the English composition has suffered. In the interests of clarity, the manuscript will require careful editing.

We thank the Reviewer for his comments. We did our best in order to improve readability of the manuscript.

Comments

52. My reading of this would lead me to believe there is direct functional evidence in the manuscript for light activated non-protonic ion channeling. The only evidence for it is from MDS. This should be made clear in the abstract.

We modified the sentence in the abstract as following: “However, the structural and functional characterization of the viral rhodopsin, together with molecular dynamics simulations, suggest that OLPVR11 might be a light-gated pentameric ion channel functionally analogous to pentameric ligand-gated ion channels.” to make it clear, that a light-gated ion channeling by OLPVR11 is our working hypothesis, but not a direct experimental evidence from the experiments and crystal structure.

142, 144, and 150. Presumably there is a mixture of oligomeric states present across all rhodopsin containing fractions. In other words it is not just monomers and pentamers that are present. This should be clarified.

Relatedly, my understanding is that the data supports the presence of pentamers but not exclusively pentamers as possibly suggested by the text.

Meaning pentamers are likely present. Explain why other oligomeric forms are observed? Are these physiologically relevant?

Taking into account two major peaks on SEC of wild type OLPVR11 we would suggest only two aggregate states with considerable protein presence – monomers and pentamers. As the pentamer assembly is highly stabilized by conservative hydrogen bonds, the appearance of other lower multimeric forms seems to be improbable. The set of multimeric forms in the cross-linking experiments appears due to the nature of the experiment. We optimized the concentration of the glutaraldehyde to have it in the amount not enough to cross all the possible sites. Therefore, on the SDS-PAGE, after pentamer denaturation, we could observe the “multimerization ladder” to

understand the exact number of the protein molecules in the multimer. However, this multimers (dimers, trimers, and tetramers) are artificial and unlikely to appear in nature. To avoid the misunderstanding we added the following text “We varied the concentration of the protein and glutaraldehyde to find the optimal condition for incomplete cross-linking of the proteins. This allowed us to observe on the SDS-PAGE the array of multimers, and using the molecular weight of the highest visible multimer we could make a conclusion about an aggregate state of the protein in a sample.” into the methods section on page 24, paragraph 3. We also observe, especially for the triple mutant, a fraction of higher aggregates, probably composed of several pentamers and/or aggregated protein, SEC allows us to eliminate most of them from the samples.

148. Indicate what stabilizes pentamer formation in the absence of pentamerization residues in the triple mutant.

We thank the reviewer for that very important comment. As we describe in the manuscript, we believe that the dense hydrogen bonding network between conservative residues at the pentamerization interface (Glu26, Arg36, Trp203) is the key determinant of the stabilization of OLPVRII pentamer. As shown by the mutational analysis, indeed, the pentamer formation is compromised significantly in the OLPVRII variant, lacking these amino acids. However, as we also notice in the manuscript on page 5: “In addition, the distorted C-terminus of one protomer interacts with the poorly ordered C-D loop of the neighboring protomer forming several hydrogen bonds between their backbone atoms and thus stabilizing the pentamer (Figure 1B, E).”. As we did not changed the C-D loop and the C-terminus of the protein, their interaction may be enough to support the existence of resting pentamer function. Moreover, we could not exclude the contribution of the hydrophobic interactions between protomers, which also plays a key role in the stabilization of OLPVRII pentamer.

As suggested by the Reviewer, we rearranged the sentences and also added the following sentence in the paragraph 2 on page 5: “The fact that triple mutant partially retained pentameric assembly shows that there are additional interactions between OLPVRII protomers, such as hydrogen bonding between C-D loop and C-terminus and hydrophobic cooperation.”.

152. Where is the evidence the protein is devoid of detergent?

For the preparation of the liposomes for the cross-linking experiments we used the detergent absorbing beads. Several consecutive applications ensure that in the presence of lipid the detergent is removed with only traces left. The formation of the liposomes was proved by DLS. To eliminate this misunderstanding we added a small comment into the methods section on page 24, paragraph 2.

363. Is it not possible to tell what the yield of protein was? Was it sufficiently low to account for the failure to detect activity or is something else amiss?

As it was suggested by another reviewer, we shifted the information about the yield of the protein into the methods section: “The total yield of the functional purified protein was approximately 2-3 mg per 5 liters of culture”, page 23, paragraph 1. This amount of protein seems to be not enough to observe the pumping activity in E.coli cells suspension.

364.”could be achieved” Were they achieved?

As we didn't observed considerable protein aggregation during the preparation of the liposomes, we suppose the concentration of the protein to be about 0.8 mg/ml. That is several magnitudes higher than in E. coli cells suspension used for direct pH measurements. To make this clear we

modified the sentence in the text as following: “Then, we measured ΔpH with the purified protein reconstituted into lipid vesicles, where high protein density was reached”.

372. Confusing. Liposomes are referred to but the opening sentence refers to BLM experiments?

We conducted BLM experiments by adsorbing proteoliposomes on the BLM. As indicated in the methods section, page 26, paragraph 2, “After the addition OLPVR II-containing proteoliposomes a photosensitivity of the samples reached maximal current amplitudes after ~90 minutes.”. To make this more clear for the reader, we added “proteo...” to the main text.

387. It would be useful to define an activity scale and what ‘weak’ corresponds to on that scale.

We thank the Reviewer for this useful suggestion. Our conclusion on the OLPVR II function as a weak proton pump is based on two independent experiments: pumping activity measurements in proteoliposomes suspension (which is a commonly-used system for testing of the microbial rhodopsins, pure from other proteins) and also BLM experiments. We observed no detectable pH changes in the proteoliposomes suspension, while the proteoliposomes with bacteriorhodopsin demonstrated notable pH changes under the same conditions (Extended Data Figure 21). However, the BLM experiments showed that OLPVR II-containing proteoliposomes are able to create photocurrents, however, much lower than that expected from a channelrhodopsin. Indeed, it was shown for channelrhodopsin 2 that its proton pumping activity is extremely weak in comparison to the ion conductance (Feldbauer et al., Proc Natl Acad Sci U S A. 2009 Jul 28;106(30):12317-22.). Taken both experiments and also a structure-based hypothesis of channel activity of OLPVR II together, we conclude that at the moment OLPVR II is shown to be a weak proton pump. It is unfortunately impossible to define an independent activity scale for the microbial rhodopsins, as one needs to perform the identical studies of all of them. We believe that such study could be of high importance and usefulness for the rhodopsins society, and indeed will deserve a separate publication.

To make it clear for the reader in the text we modified the last paragraph of the ‘Pumping activity of OLPVR II’ section as following: ‘Taking into account the arrangement of functionally important amino acids in the protein and the results of the BLM experiments, we conclude that, at least in liposomes, OLPVR II acts as an outward proton pump. However, the pumping activity is weak in comparison with expected channel conductance by considering one photocycle. Indeed, it was shown that ChR2 also possesses outward proton pumping activity, which is negligible compared to its ion channeling⁴⁰.’

399. A pore that is ~4 Å across is big enough to allow water and other substances to pass. This mean the pentamer creates an open pore in the membrane? Blockage of polar substances is proposed to occur by way of the hydrophobic gate. Does this mean the pentamer provides access to small hydrophobic substances? What might be the consequences if so?

We thank the Reviewer for these interesting questions. As written in our manuscript in paragraph 4 on page 12: “Calculation of pore radii shows a constriction in this region, with a minimum of 1.93 Å (Figure 6C)”. Taking into account that size of the water molecule is ~3 Å, water molecules and small ions could not pass through the pore, created by the OLPVR II pentamer. Indeed, as it is known for many biological pores with diameter smaller than 10 Å, the hydrophobic gating mechanism could already block the transport of any ions (see, for example, the following review Aryal et al., J Mol Biol. 2015 Jan 16;427(1):121-30). As it correctly noticed by the Reviewer, the hydrophobic gating mechanism does not allow the passage of polar substances through the OLPVR II central pore, due to hydrophobic constriction near Phe24 and Leu28 residues.

On the other hand, hydrophobic and/or amphiphilic molecules can bind in the central pore of OLPVR II. Indeed, in our structure we observed the hydrocarbon chain in the pore. The detergent

molecules were found in the central pores of pentameric ligand-gated ion channels (pLGICs) (Sauguet et al., EMBO J. 2013 Mar 6;32(5):728-41). We believe that in case of OLPVRII this is an artifact of the crystallization and the main function of the protein is ion channeling. At the same time, we could not exclude the translocation of hydrophobic/amphiphilic molecules as a putative biological function of the viral rhodopsin. The role and consequences of this transportation remain unclear and could not be elucidated at the moment. This hypothesis, together with the proposed function of OLPVRII as a pentameric light-gated ion channel require further comprehensive studies, which, as we believe, will result in many interesting works and will contribute to the understanding of the virus-host interactions.

422. How active is the channel expected to be and how does this activity compare to other channels in current optogenetics use? In other words, how realistic is the idea of using this system for optogenetics applications?

Our simulations demonstrate that our crystal structure exhibits an anion-selective pore with a closed hydrophobic gate, and that only small conformational changes, such as rotation of hydrophobic gate residues F24 and L28 away from the pore center, might be sufficient to functionally open the pore (Figure 6). Consistent with the proposed hydrophobic gating mechanism, we observe frequent spontaneous chloride permeation events in equilibrium MD simulations of the F24A/L28A mutant (i.e. in absence of applied transmembrane voltage) on the microsecond simulation timescale, equivalent to what has been observed for pentameric ligand-gated ion channels (e.g. Sauguet et al., EMBO J. 2013 Mar 6;32(5):728-41). These observations suggest significant chloride permeability similar to other established ion channels.

However, in absence of an open-state crystal structure, we refrained from investigating ion permeation at a more quantitative level. In particular, single-channel conductances determined from MD simulations are expected to be particularly sensitive to the precise geometry of the open ion pore, which is still unknown. Thus, although exact quantification of these pore properties awaits future research, our current simulations robustly support the existence of anion-selective permeation pathway in OLPVRII.

710. Spell out what DDM is. Is it alpha or beta DDM or a mix?

n-Dodecyl- β -D-Maltopyranoside. Corrected.

715. Rewrite for clarity.

We rewrote the sentence as was proposed by the reviewer.

726. Did the ratio change across the chromatogram from monomers to pentamers?

After optimization of the purification protocol, we obtain the monomeric fraction of the protein with the peak ratio about 1.0-1.3 and pentameric fraction with the peak ratio of about 1.3-1.7. Generally, peak ratio of the pentameric fraction is slightly higher due to the aggregates impurities.

736. State lipid concentration.

The amounts of lipids and buffer were added.

739. Provide more details regarding detergent removal with beads.

The details were added.

741. State the type and size of liposomes that result and were used. Multilamellar, unilamellar, a mix?

We had unilamellar vesicles with the size 100-300 nm depending on lipid composition. This information is added to the Materials and Methods section.

750. Report source of lipoprotein and conditions for nanodisc formation.

We indicated in text that MSP was expressed and purified in-house. The protocol of the nanodisc preparation reproduced the published protocol (Ritchie et al., Methods in Enzymology 464, 211–231 (2009)).

755. PS contains an amino group.

We thank the Reviewer for this important question. We suggest that may be the decreased concentration of the amino group in used lipid mixture allowed us to obtain the clear picture of the pentamer formation. E. coli polar lipids extract contains about 70% of PE lipids. We underlined this fact in the main text: “For crosslinking experiments liposomes from E. coli polar lipids turned out to be not suitable, as a natural mixture of lipids contains considerable amount of NH₂-groups that block the crosslinking reaction.”

756. mole or weight ratio?

This is a weight ratio, and we added the information into the main text.

784. Report the concentration of cells and liposomes used in light-induced pH change studies

On page 25, paragraph 2 it is indicated that “After that the cells were resuspended in 100 mM NaCl solution and adjusted to an OD₆₀₀ of 8.5.” We added the information concerning the density of the liposomes on page 23, paragraph 2.

819. State how much shorter.

The photocycle of the OLPVRII monomers in the detergent had the length of 35 ms comparing to 70-100 ms long photocycles in the other environments. We added this information to the manuscript.

821. Rewrite for clarity.

We improved the readability of the sentence. “Close properties of photocycles measured for a membrane protein in different environments are not unique to OLPVRII.”

849. Link does not work. Provide information requested.

We exchanged the link with the requested parameters. “Then we spotted 150 nl drops of mesophase on a 96-well LCP glass sandwich set (Paul Marienfeld GmbH, Germany; bottom slide is covered with a 0.2 mm high spacer that has 96 recesses of 5 mm diameter) and overlaid them with 400 nL of precipitant solution using NT8 crystallization robot (Formulatrix).”

901. Explain how cavity was determined.

The cavity was determined using HOLLOW software (Ho & Gruswitz, BMC Struct. Biol. 8, (2008)). The script accurately identifies the cavities inside the protein and also on its surface. By default, the program analyze the surface at maximal distance of 7 Å from the protein, which is usually enough for obtaining the inner cavities and relatively large pores in the protein. To calculate accurately the profile of the central pore and due to its notable width, we increased the maximal distance from the protein to 25 Å. We added this information to the Materials and Methods section and also to the Figure 1 legend.

905. No certainty the fragment is lipid. It could be detergent. Suggest using hydrocarbon.

We thank the Reviewer for this notice. We changed the figure legend and the Figure 2 as suggest by the Reviewer.

Fig 3. Explain pink volumes in this and other figures.

The pink volumes represent the cavities inside the proteins, calculated using HOLLOW, as state in the Material and methods section. We modified the legends of Figures 3,4 and Extended Data Figures 11, 14-17 to explain clearly what the pink volumes are as following: "Cavities inside the protein protomer are colored pink."

907. Brown arrows represent speculation and should be identified as such.

To make it clear that the brown arrows represent only working hypothesis, we modified the sentence from the legend of the Figure 4 as following: "The brown arrows show the putative sequence of structural rearrangements transduced from RSB to the pore interface."

Edf 6. Explain the large shift in elution volume between the wt and the triple mutant.

The relative shift of the peaks in SEC can be caused by multiple reasons or their combination, we do not know the exact answer. Possible reasons could include the mutual influence of the positions of the peaks and their heights on each other, the influence of the mutation itself, the difference of the detergent binding to the wild type and mutant proteins, differences in mobility of the samples, and different setups where the experiments were carried out. For not to be too speculative, we will refrain from the comments in the main text as, in our opinion, here the SEC profiles shows very clear the qualitative picture of the monomer:pentamer distribution, but they are not suitable for quantitative conclusions. However, we included a comment into the figure legend: "We suppose that the relative shifts of the monomers and pentamers are probably caused by a combination of several reasons, including the mutual influence of the positions of the peaks and their heights on each other, the influence of the mutation itself, the difference of the detergent binding to the wild type and mutant proteins, differences in mobility of the samples, and different setups where the experiments were carried out. Nevertheless the elution profiles clearly illustrate the distribution of the protein between monomeric and pentameric fractions."

945. State how pKa is determined.

We modified the figure legend for clarity and added the information concerning the determination of the pKa of the proton acceptor and Schiff base to the corresponding part of the Materials and Methods on page 25, paragraph 2: "Fitting at lower pH values of the dependence of the retinal absorbance maximum on the pH value with sigmoidal curve gave the pKa of the proton acceptor. Fitting at higher pH values of dependence of the absorbance at 514 nm on the pH value with sigmoidal curve gave the pKa of the Schiff base."

952. Are these two pumping experiments directly comparable? Same liposome density, same protein density, etc? This should be clarified in the legend.

Yes, the experiments are directly comparable. The liposome density, protein concentration, lipid composition, buffer and setup were the same in two cases. We added the following text: "As a positive control we used proteoliposomes with HsbR that was expressed and purified as described²¹." on the page 25, paragraph 1 and indicated that the experiments are directly comparable in the figure legend.

Edf 22/23 legends mixed up.

We changed the order of the figures.

Edf 24. Does the mechanism of channel opening only require one retinal to isomerize as suggested in the scheme? What is the likelihood of multiple isomerizations?

First of all, we would like to thank the Reviewer for the comments on the Extended Data Figure 24. We also would like to stress, the Extended Data Figure 24 is only a putative mechanistic scheme to make clear our working hypothesis that OLPVR II serves as a light-gated pentameric anion channel and explain which structural rearrangements could lie beneath its activation by light. As a result of that, all the ideas, reflected in the Extended Data Figure 24 are speculations and requires further comprehensive investigations by various methods. We believe, that this experiments will be performed in the nearest future and will lead to numerous of publications on OLPVR II and other viral rhodopsins.

To make clear, that this figure is only a hypothetic model, we changed the Extended Data Figure name to "A putative mechanistic scheme of OLPVR II opening/activation".

Regarding the multiple retinal isomerizations, we do not know. Two different possibilities cannot be excluded: cooperative and non-cooperative isomerization of the retinals within pentamer. However, it is a question for future studies. Still, we expect that efficiency of cooperative isomerization could be higher at lower light intensities, and therefore we would not expect non-cooperative way of retinals isomerization and channel opening.

Why only chloride channeling? Are there other possibilities? Water and protons can cross. Proton movement would collapse the membrane gradient.

Our simulations of membrane-embedded OLPVR II have been performed in aqueous NaCl solution, and we thus monitored water, Na⁺, and Cl⁻ permeation. Simulations of the wild-type protein reveals a closed state without any ion permeation, and simulations of F24A/L28A OLPVR II (to mimic the open hydrophobic gate) show increased water permeability and frequent spontaneous Cl⁻ permeation events, but no Na⁺ permeation (Figure 6). Adding the R29E substitution, we observe frequent Na⁺ permeation events in our simulations, without any Cl⁻ permeation (Extended Data Figure 23). These findings demonstrate that R29 dominates the anion selectivity of OLPVR II, and that charge inversion of R29 can alter the anion/cation selectivity.

In this study, we decided not to investigate permeabilities for other anions than Cl⁻, since relative anion permeabilities are expected to be much more sensitive to the precise pore geometry, than relative anion/cation permeabilities. In summary, our simulations provide strong support for an anion-selective permeation pathway in OLPVR II, but quantitative determination of relative anion selectivities awaits future research (including determination of an open state structure).

How confident are you about the directionality of flow?

We performed equilibrium simulations wild-type and mutated OLPVRII (i.e. in absence of transmembrane voltages). In simulations of L24A/F28A OLPVRII, we observed frequent, spontaneous Cl permeation events in both inward and outward direction, reflecting non-directional diffusive motion of Cl ions in the open pore. These simulations suggest that Cl conduction is possible in both directions; however, we refrain from quantitative predictions regarding ion current rectification. This property critically depends on the precise pore geometry and will be addressed in the future as soon as an experimental open-state structure becomes available.

Is there evidence that isomerization can trigger helix A rotation? Was this/can it be modelled by MDS? Would this rotation open the hole?

Structural vicinity of the all-trans retinal bound to K195 to helices A and B suggests that conformational changes of the retinal may induce slight tilting or rotation of these helices. Furthermore, the hydrophobic gating mechanism predicts that only small conformational changes, that move the sidechains F24 and L28 away from the pore center to reduce the effective hydrophobicity, may be necessary to functionally open the ion conduction pathway. Similar mechanisms have also been found in ligand-gated ion channels (Zhu & Hummer, Biophys. J. 103, 219–227 (2012)).

All our present OLPVRII simulations have been performed with the retinal in the all-trans state so far, and we consider computational studies on light-induced conformational changes outside the scope of the current manuscript, since approaches for appropriate experimental validation first need to be developed. Investigation of the trans-to-cis isomerization and the mechanisms of photoactivation in OLPVRII will be important aims in a future project that will include experimental determination of the open-state structure of OLPVRII to assess these conformational changes.

What function would be served by a light driven transport of the type proposed?

We thank the Reviewer for that important and difficult question. As we discuss in the Concluding remarks section of the manuscript, we believe, that anion (chloride) light-gated channels may modulate the ion homeostasis of cells, infected by the virus, which can be crucial for virus reproduction. Another possibility is that these channels may influence the phototaxis of the infected cells, in the same manner as channelrhodopsins of Chlamydomonas reinhardtii serve as its receptors for phototaxis and the photophobic responses. However, these hypotheses require further investigations.

Label sides of membrane.

We labeled the sides of the membrane on the Extended Data Figure 24.

Reviewer #4 (Remarks to the Author):

The authors addressed my critique satisfactorily. Especially they addressed the monomer/pentamer state of the rhodopsin, a concern that was also raised by other reviewers.

We thank the Reviewer for the comments on our manuscript. We are happy that we addressed all the questions of the reviewer.

Reviewers' Comments:

Reviewer #3:

Remarks to the Author:

Revised version 2 of the manuscript is much improved. However, the English composition remains problematic in parts. In the interests of clarity, the manuscript will require careful editing.

Comments

38. It should be stated clearly in the summary that no experimental evidence is provided in this study to show the protein is a non-protonic ion channel.

53. This statement is ambiguous. Keep in mind that a proton is an ion.

149. 'devoid of detergent' is too strong a statement unless separate analysis shows the sample is truly free of detergent.

155. Change 'shows' to 'suggests'

368. The implication is that the protein has relatively low activity. This should be stressed in the manuscript since it may impact on potential downstream applications.

391. Confusing. Rewrite this sentence for clarity.

392. Confusing. Rewrite for clarity. A proton is an ion.

735. "After optimization of the purification protocol, we obtain the monomeric fraction of the protein with the peak ratio about 1.0-1.3 and pentameric fraction with the peak ratio of about 1.3-1.7. Generally, peak ratio of the pentameric fraction is slightly higher due to the aggregates impurities."

This information should be included in Methods.

749. This statement can only be made if follow up analysis shows the protein is free of detergent.

754. State the evidence that these are unilamellar. This is important to know when comparing activities.

779. Confusing sentence. Rewrite for clarity.

842. It is unclear what 'close properties' means. Rewrite for clarity.

EDF6. The SEC should be calibrated to correct for the 'different setups'. If not, this should be stated in the Methods and legend.

It is unclear what 'mutual influence' means. It should be rewritten for clarity.

'different mobility of the samples' This is just restatement of the problem. It is not an explanation and should be rewritten or removed.

983. The mechanism suggests the protein operates as a dimer. Clarification is required.

Reviewer #3 (Remarks to the Author):

Revised version 2 of the manuscript is much improved. However, the English composition remains problematic in parts. In the interests of clarity, the manuscript will require careful editing.

We would like to thank the reviewer for the work and helpful comments on our manuscript.

Comments:

38. It should be stated clearly in the summary that no experimental evidence is provided in this study to show the protein is a non-protonic ion channel.

We added to the abstract the information that structural and functional data and molecular dynamics suggest that OLPVR_{II} might be a light-gated pentameric ion channel, but future patch clamp experiments should prove the suggested channel activity directly.

53. This statement is ambiguous. Keep in mind that a proton is an ion.

During the charge transfer by ion channels the protons are also transferred. We do not see any ambiguity in this sentence.

149. 'devoid of detergent' is too strong a statement unless separate analysis shows the sample is truly free of detergent.

We reformulated this part of the sentence as “purified in detergent monomers, when placed into lipid environment, assemble into the pentamers again”.

155. Change 'shows' to 'suggests'

Replaced.

368. The implication is that the protein has relatively low activity. This should be stressed in the manuscript since it may impact on potential downstream applications.

We reformulated the sentence as follows: “However, the charge transfer per photocycle is small compared to that of the expected channel activity.” We think that now it reflects better what we wanted to discuss in this paragraph. Indeed, the relative activity is hard to discuss, because, on the one hand, OLPVR_{II} showed the comparable to proteorhodopsin photocurrents (see ref. 38 in the manuscript), but, on the other hand, BLM experiments can show considerable variation of the signal depending on liposome adsorption and orientation of the proteins in lipid bilayer (both parameters are difficult to control). Also an equal amount of incident photons could lead to different currents for different proteins due to the differences in absorption and quantum yield. For OLPVR_{II}, we suppose that one proton is translocated per photocycle, but the protein could turn out to be the leaky protein pump similar to Chr2, and thus the signal could decrease. Therefore, we suppose to compare the charge transfer of protein pumping activity and channeling activity that are evidently different.

391. Confusing. Rewrite this sentence for clarity.

392. Confusing. Rewrite for clarity. A proton is an ion.

391 and 392. *We modified the first sentence to improve its readability. We consider that the proton is an ion, however, per photocycle, the charge transfer of proton pumping activity is small compared to the charge transfer of expected channeling activity, when protons and other ions are transported. This statement holds true for ChR2 that was demonstrated in Ref. 42.*

735. “After optimization of the purification protocol, we obtain the monomeric fraction of the protein with the peak ratio about 1.0-1.3 and pentameric fraction with the peak ratio of about 1.3-1.7. Generally, peak ratio of the pentameric fraction is slightly higher due to the aggregates impurities.”

This information should be included in Methods.

We modified Paragraph 1 on Page 17 accordingly.

749. This statement can only be made if follow up analysis shows the protein is free of detergent.

We changed “complete” to “ample”.

754. State the evidence that these are unilamellar. This is important to know when comparing activities.

We prepared the liposomes using detergent and sonication. Then, when liposomes were ready, we sometimes checked the size of the liposomes with DLS and received the values mostly less than 200 nm. Generally, multilamellar liposomes are considerably bigger than unilamellar. This fact allows us to suppose that the liposomes obtained were unilamellar. However, we have not conducted direct experiments to prove that we obtained the unilamellar liposomes. Nevertheless, we would suppose to keep the word “unilamellar”.

779. Confusing sentence. Rewrite for clarity.

We improved the readability of the sentence.

842. It is unclear what 'close properties' means. Rewrite for clarity.

We modified the sentence and removed the phrase “close properties”.

EDF6. The SEC should be calibrated to correct for the ‘different setups’. If not, this should be stated in the Methods and legend.

It is unclear what 'mutual influence' means. It should be rewritten for clarity.

'different mobility of the samples' This is just restatement of the problem. It is not an explanation and should be rewritten or removed.

We modified the legend of Supplementary Figure 6 and Methods (Paragraph 1 on Page 17).

983. The mechanism suggests the protein operates as a dimer. Clarification is required.

We modified the figure legend to eliminate any misunderstanding.